# BERT-Sort: A Zero-shot MLM Semantic Encoder on Ordinal Features for AutoML

**Mehdi Bahrami   Wei-Peng Chen   Lei Liu   Mukul Prasad**

Fujitsu Research of America, Sunnyvale, CA

**Abstract**   Data pre-processing is one of the key steps in creating machine learning pipelines for tabular data. One of the common data pre-processing operations implemented in AutoML systems is to encode categorical features as numerical features. Typically, this is implemented using a simple alphabetical sort on the categorical values, using functions such as *OrdinalEncoder*, *LabelEncoder* in *Scikit-Learn* and *H2O*. However, often there exist semantic ordinal relationships among the categorical values, such as: quality level (i.e., ['very good' > 'good' > 'normal'> 'poor']), or month (i.e., ['Jan'< 'Feb' < 'Mar']). Such semantic relationships are not exploited by previous AutoML approaches. In this paper, we introduce BERT-Sort, a novel approach to semantically encode ordinal categorical values via zero-shot Masked Language Models (MLM) and apply it to AutoML for tabular data. We created a new benchmark of 42 features from 10 public data sets for sorting categorical ordinal values for the first time, where BERT-Sort significantly improves semantic encoding of ordinal values in comparison to the existing approaches with 27% improvement. We perform a comprehensive evaluation of BERT-Sort on different public MLMs, such as RoBERTa, XLM and DistilBERT. We also compare the performance of raw data sets against encoded data sets through BERT-Sort in different AutoML platforms including AutoGluon, FLAML, H2O, and MLJAR to evaluate the proposed approach in an end-to-end scenario, where BERT-Sort achieved a performance close to a hard encoded feature. The artifacts of BERT-Sort is available at https://github.com/marscod/BERT-Sort.

## 1 Introduction

An Automated-Machine Learning (AutoML) platform aims to automate the process of feature engineering, data engineering, hyper-parameter optimization, training, prediction, and deployment, where it minimizes human supervision in all stages (He et al., 2021). Each data set may contain a variety of data types including ordinal values where the order of values is important. Let $C_1 \prec C_2 \prec ... \prec C_n$ denote a fixed set of arbitrary classes of $C$. For instance, *UCI Audiology data set* (Porter, 2019) includes a feature of *Air* with a set of unique values of *Normal ≺ Mild ≺ Moderate ≺ severe ≺ profound*. Although this field is not a target feature, the order of the values carries semantic meaning. AutoML platforms encode each feature based on their types and content of values. Often AutoMLs encode categorical features as an integer array (Pedregosa et al., 2011) function. For instance, *H2O AutoML* (LeDell et al., 2018) and *Scikit-Learn* use *categorical_encoding* and *OrdinalEncoder* (SkL, 2021) functions, respectively, to transform categorical values to an integer array. However, all categorical encoders rely on an alphabetical sort function such as *Numpy Sort* (num, 2021) where the encoded values are based on the sorting results of categorical values alphabetically (Oliphant, 2006) (LeDell et al., 2018). Such a simple method may fail to capture the semantic relationships between values. For instance, the value *'profound'* in the *Air* feature of *UCI Audiology* data set represents more serious level than the value *'severe'* but *OrdinalEncoder* returns the opposite result. Similarly, alphabetically sorting values of *num-of-cylinders* feature in *UCI Automobile Data Set* returns *['eight' ≺ 'five' ≺ 'four' ≺ 'six' ≺ 'three' ≺ 'twelve' ≺ 'two']* as $[C_1 \prec C_2 \prec ... \prec C_7]$, respectively, where the ordinal values have been misplaced. As a result, such

incorrectly encoded values can pose more challenges to any machine-learning algorithms to predict the target value of *price* based on increase/decrease value of *cylinders*.

One hypothesis to be verified in this paper is that AutoML platforms incorrectly encoding ordinal categorical features might result in degraded performance. To address this issue, we propose a novel approach *BERT-Sort* which utilizes pre-trained Masked Language Models (MLM) (Devlin et al., 2018) in a zero-sot setting to semantically sort and encode ordinal values. The following are our main contributions in this study.

(i) A zero-shot systematic sorting algorithm to sort ordinal values is introduced in Section 3;

(ii) We compose a benchmark of 10 real-world data sets with 42 ordinal features for the first time, which is explained in Section 4 (detail in Appendix A);

(iii) We conduct a comprehensive performance evaluation of benchmarks by i) comparing the results of BERT-Sort (with initialization on 4 different publicly available MLMs) and OrdinalEncoder (which is widely used in different AutoML platforms), and ii) evaluating between raw data set and encoded data set through BERT-Sort in 4 different AutoML platforms of AutoGluon, FLAML, H2O, MLJAR in Section 4.

## 2 Related Works

In the supervised approach, researchers and practitioners utilize existing limited data sets with ordinal values to develop a model where it can encode ordinal values and predict unseen ordinal values for encoding purposes. However, the adaptation of a trained model from one domain to another is extremely limited. Therefore, most related works in the supervised approach can be used in a set of selective domains/languages and it can be used in a form of training a model for a particular domain. An early study by McCullagh (1980) and later other studies by Christensen (2015) and Harrell (2015), introduced a general class of regression models for ordinal values where it utilizes the ordinal values through various modes of stochastic ordering. In another recent study by Lausser et al. (2020), the authors developed an ordinal subcascades detection and encoding process, but the authors mentioned that their works have a limitation where analyzing suitable data representations may give a better answer for ordinal encoding. Dahouda and Joe (2021) proposes a deep-learned embedding technique for categorical features encoding. The proposed technique is a distributed representation for categorical features where each category is mapped to a distinct vector, and the properties of the vector are learned while training a neural network. In our proposed approach, we utilize the semantic understanding of MLMs to overcome the supervision and the limitation of ordinal subcascade encoding and other similar approaches.

In addition to the aforementioned studies, categorical encoders have also been widely applied in AutoML platforms. Auto-sklearn provides *ordinal encoding* or *one-hot encoding* as choices for categorical features (Feurer et al., 2020). MLJAR (mlj, 2022) (Płońska and Płoński, 2021) converts categorical features into numeric with *label encoder*, *one-hot encoder* or *target encoder*, which is automatically selected based on feature cardinality and AutoML training stage (mlj, 2022). As explained by LeDell et al. (2018), H2O utilizes tree-based models (Gradient Boosting Machines, Random Forests) to support group-splits on categorical variables, so categorical data can be handled natively. However, as explained by Zhou and Hooker (2021), the default split-improvement method is biased towards increasing the importance of features with more potential splits especially when we are dealing with a large number of ordinal values. H2O also uses *categorical_encoding*, which specifies the encoding scheme to use for handling categorical features. In AutoGluon, each categorical feature is mapped to monotonically increasing integers (Aut, 2022a). *AutoKeras* defines an argument *categorical_encoding*, which specifies whether to encode the categorical features to numerical features (Aut, 2022b). However, these categorical encoders rely on purely alphabetically sort functions such as *Numpy Sort*(num, 2021) where it rearranges the values by alphabetically

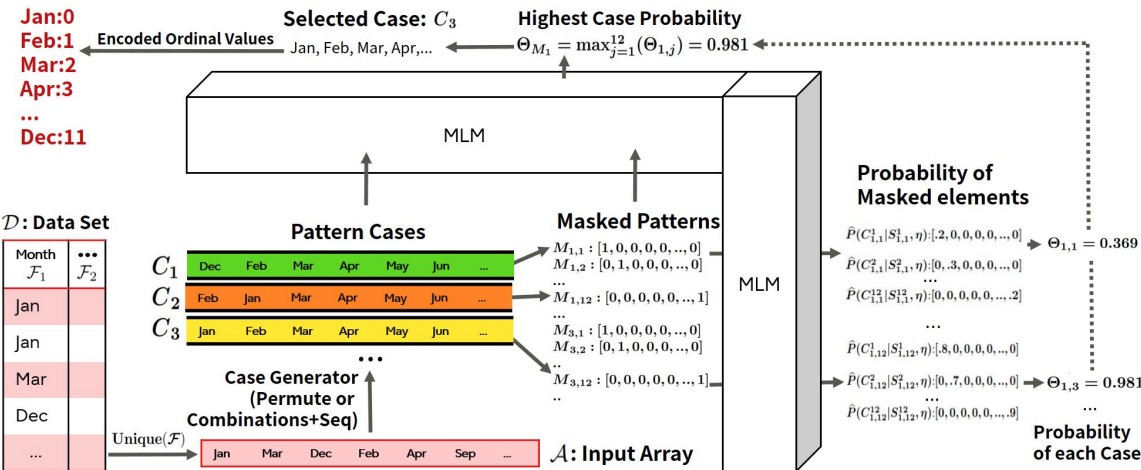

Figure 1: An overall process of BERT-Sort approach

sorting values, then it assigns an integer value based on the index of sorted values (Oliphant, 2006) (LeDell et al., 2018). Although the method can encode the categorical data sets, it fails to encode ordinal values by misplacing the values in incorrect orders. In this paper, we introduce an approach that takes advantage of ordinal values, their semantic definitions to define an approach that can be universally applied toward diverse domains/languages without any supervision.

## 3 BERT-Sort

The basic idea of BERT-Sort is to utilize the power of a pre-trained language model to recognize the order of a given sequence of values. Language models learn from a large number of text corpus where a language model processes the context of sentences and paragraphs. Masked-Language Modeling (MLM) is a self-supervised learning of text representations where it computes the probability distribution of a masked token from a given context. A masked language model can be used across different down-stream tasks such as text generation. Each feature of a given data set may contain categorical ordinal values. The tokens of ranked ordinal values could be seen in many documents. For instance, we may find a large number of documents that includes *'February'* appearing after *'January'*. In NLP down-stream tasks, such as text generation, we may replace a subset of tokens of a given input string with a distinctive character of *[MASK]* and the objective is predicting the masked token. We utilize this unmasking process to find the orders of a set of given values based on their probability of orders according to a pre-trained MLM.

### 3.1 Problem Formulation

Figure 1 shows the overall process of the proposed approach, BERT-Sort, where it aims to find the best order of ordinal values by computing the maximum probability of appearances of values in a set of possible orders. First, BERT-Sort captures all categorical features ($\mathcal{F}$) of a given data set, $\mathcal{D}$. Let $\mathcal{A}_i$ denotes all unique values of $i$th feature ($\mathcal{F}_i$). If $||\mathcal{A}_i|| \leq \varphi$, where $\varphi$ is a threshold of the maximum number of unique values, BERT-Sort applies to $\mathcal{A}_i$ (a candidate for ordinal feature). In Section 4, we extend this rule-based approach to automate the detection of ordinal values by applying BERT-Sort to detect if there is an ordinal relationship between values. BERT-Sort generates $N$ different possible permutation cases (different orders of ordinal values) from $\mathcal{A}_i$, which is denoted as $\mathcal{C}_{i,j}$ where $j = [1, .., N]$. $\mathcal{C}_{i,j}^k$ denotes $k$th element of $\mathcal{C}_{i,j}$. Since it is computationally expensive to generate all possible permutations for a large number of elements, we explain a sequential approach in Section 3.2 to process a fewer number of cases. BERT-Sort produces $k$ different masked patterns for $i$th feature, and $j$th case as $M_{i,j}^k$ where $k = 1..||\mathcal{A}_i||$. In each iteration ($k$) a single value of $\mathcal{C}_{i,j}^k$ is

masked. Let $\mathcal{S}_{i,j}^k$ denotes a generated sentence by applying each masked pattern of $M_{i,j}^k$ to $\mathcal{C}_{i,j}^k$ where it is masked $k$th element of $\mathcal{C}_{i,j}$. For instance, for a given three elements ($k = [1, 2, 3]$) of $\mathcal{A}_i[A, B, C]$, it applies $M_{1,1}^1 = [1, 0, 0]$, and it generates a sentence of $\mathcal{S}_{1,1}^1 = $ [CLS][MASK], B,C.[EOS]. In this step, different sentence structures can be generated (i.e., replacing the *comma* between the ordinal values with *a blank space*). We explain the performance of building different sentence structures in Section 10.

Next, BERT-Sort performs a model inference on initiated MLM (i.e., RoBERTa) to unmask $\mathcal{S}_{i,j}^k$ and MLM returns $\eta$ number of retrieved tokens, which is denoted by $\mathcal{W}_\eta$. if $\mathcal{C}_{i,j}^k \in \mathcal{W}_\eta$, it indicates that MLM returns a probability for the sequence of $C_{i,j}^k$ with the context of $\mathcal{S}_{i,j}^k$ and it is denoted by $\widehat{P}(C_{i,j}^k|S_{i,j}^k, \eta)$; otherwise it returns 0. Finally, it finds the average probability of all masked sentences of $i$th feature for $j$th case ($\Theta_{i,j}$) as follows.

$$\Theta_{i,j} = \begin{cases} \frac{\sum\limits_{k=1}^{||\mathcal{A}_i||} \widehat{P}(\mathcal{C}_{i,j}^k|S_{i,j}^k,\eta)}{||\mathcal{A}_i||}, & \text{if } \mathcal{C}_{i,j}^k \in \mathcal{W}_\eta \\ 0, & \text{otherwise} \end{cases} \quad (1)$$

$\Theta_{i,j}$ denotes the score of appearing a sentence (which is constructed from ordinal values) for $i$th feature, and $j$th order (case). $\Theta_{M_i} = \max_{j=1}^{||A||}(\Theta_{i,j})$ which is selecting a case with the highest score (the best order or sequences of input elements). Figure 3 shows an example of score computation for 3 elements. Intuitively, BERT-Sort finds the most likely correct order from all possible permutations with utilizing a pre-trained MLM. Different MLMs can be applied under the same architecture across different domains (i.e., medical domain) and languages (i.e., Chinese). We explain the boarder impact of BERT-Sort in Section 12.

## 3.2 Handling a large number of ordinal values

Although BERT-Sort semantically ranks values instead of non-semantic approaches (i.e., alphabetical sort), it is computationally intractable for BERT-Sort to directly support a large permutations of ordinal values. BERT-Sort applies two approaches for handling a large number of ordinal values.

**First**, BERT-Sort uses a divide and conquer approach for sorting elements if there are any repeated words among ordinal values. Let $\mathcal{C}_i = [W_1^i, .., W_n^i]$ denotes n-gram (Brown et al., 1992) composition of $i$th ordinal value. The following shows five steps for this process. i) It generates a set of groups where each group has a word (largest n-gram words) in common, and it denotes common words. (i.e., define a group of *[Lava Hot, Boiling Hot, Hot]*). Note that in this step, if $W_m^i$ of $\mathcal{C}_i$ is selected for a group, then $[W_1^i, .., W_n^i]$ will be added into the group, and it will not repeat the process on other words of $[W_1^i..W_{m-1}^i, W_{m+1}^i..W_n^i]$; ii) It selects a group leader word where it is most frequent largest n-gram word among the group values (i.e., *'Hot'* is selected as group leader in previous example); iii) It sorts elements within each group. (i.e., *[Hot ≺ Boiling Hot ≺ Lava Hot]*); iv) It sorts elements of **group leaders** and **unique values** (i.e., *[Cold ≺ Hot]*); Finally, v) it replaces each sorted group leaders with their sorted values of the group (i.e., *[Cold ≺ Hot ≺ Boiling Hot ≺ Lava Hot]*).

**Second**, BERT-Sort uses a sequential adding procedure by sorting $\zeta$ number of elements, then adds the rest of elements sequentially. It aims to avoid generating a large number of permutation cases when $\zeta < ||\mathcal{A}||$. Figure 2 shows an example of sequential sorting values for 12 ordinal value elements (months abbreviations). BERT-Sort uses a sequential approach in three steps as follows. i) It finds the best candidate for $\zeta$ blank spots (in the figure, it is initiated with $\zeta = 5$ spots); ii) Once it finds the best candidates for $\zeta$ blank spots as initial sorted elements (it finds *[Jan ≺ Feb ≺ Apr ≺ Jun ≺ Sep]* for 5 spots in the example), then it adds each remaining element to the initial sorted elements (in the second iteration, it finds the best position for *"Oct"* in initial sorted 5 elements); Finally, iii) it repeats step (ii) until all elements are added to the final sorted values. This process

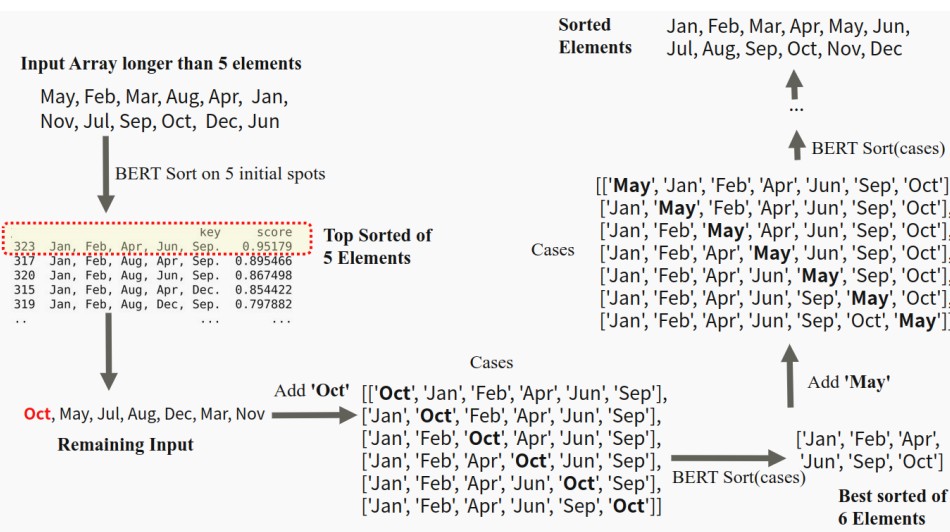

Figure 2: An example of sequential sorting elements for large number of ordinal values where $\zeta = 5$

reduces the computation time from $\mathcal{O}(n!)$ where $n = ||\mathcal{A}||$ to i) a combination function for selecting $\zeta$ ordinal values from $n$ number of values with $\mathcal{O}(n^{\zeta})$, and ii) a sequential adding function with $\mathcal{O}(n^2)$ where the whole procedure can be completed in $\mathcal{O}(n^{\zeta} + n^2)$ s.t. $2 < \zeta < \frac{n}{2}$.

BERT-Sort aims to have a better accuracy with a higher probability score with lengthy contents (larger $\zeta$ have better accuracy, see Section 11). For instance, $\Theta_M$ with $U([MASK]) = Jan$ where $U(c) = t$ returns the probability of a given masked context of $c$ if $t$ token exist in the list and considering $c =$ "[CLS][MASK], Feb, Mar, Apr.[EOS]" is higher than $U([MASK]) = Jan$ where $c =$ "[CLS][MASK], Feb.[EOS]". Therefore, we recommend to keep maximum possible of $\zeta$ elements per available computation resources. For instance, $\zeta = 5$ if $||\mathcal{A}|| \leq 12$, and $\zeta = 3$ if $12 < ||\mathcal{A}|| \leq 20$. Algorithm 1 shows the details of BERT-Sort procedures.

## 4 Experiments

**Experimental Setup**. We evaluate the proposed BERT-Sort approach under two cases: i) the performance of BERT-Sort in detecting and encoding categorical ordinal features; ii) the effectiveness of encoding applied to categorical ordinal features by BERT-Sort before input to various AutoMLs.

We use 10 different publicly available real-world data sets because it includes categorical ordinal values. We annotate and generate 42 different categorical features (as explained in Appendix A), where we compare the performance of Scikit-Learn OrdinalEncoder as a baseline against BERT-Sort encoder. We initiate BERT-Sort algorithm with 4 different popular and publicly available MLMs: DistilBERT, RoBERTa, XLM-RoBERTa and BERT-base-uncased ($M_{1..4}$ respectively) in a zero-shot setting. The inference on MLM can be optimized on CPU to reach 2ms per case (Philipp Schmid, 2022) and further detail explained in Section 8. We compare the results of different MLMs and recommend the best MLM to researchers and practitioners. The details of MLM, configurations and BERT-Sort hyper-parameters with a link to reproduction are presented in Section 7 and Section 8.

Like other encoders, such as *OrdinalEncoder*, *LabelEncoder*, we use BERT-Sort to rank ordinal values semantically (Altınel and Ganiz, 2018), then assign an integer for each element per their orders. Since the orders of the values are principal factors in either ascending or descending, we do not distinguish between two ranks. However, we expect that BERT-Sort returns ordinal values ranked in ascending order (i.,e., 'low' to 'high' or 'Jan' to 'Dec') because most documents (which has been used to train MLMs) are written in ascending format. For instance, we may find many documents in Wikipedia that indicate 'Jan, Feb' and fewer document that include 'Feb, Jan'.

**Algorithm 1** BERT-Sort Procedures

---

1: **procedure** BERT_SORT($\mathcal{D}, \zeta$)
    ▷ Sorts each feature, $\mathcal{F}$, of input data set, $\mathcal{D}$ with a given $\zeta$ parameter.
2:    **for** $\mathcal{F}$ in $\mathcal{D}$ **do**                                                   ▷ process each feature of given data set
3:       **for** $\mathcal{U}_f$ in Unique($\mathcal{F}$) **do**                               ▷ capture pre-processed unique values
4:          $\mathcal{R}(\mathcal{U}_f) \leftarrow$ G($repeated\_words$) **if** $\exists g$ **else** $\mathcal{U}_f$     ▷ grouping repeated words among unique values;
5:                                                      ▷ otherwise generate a single group
6:          **for** $\mathcal{G}$ in $\mathcal{R}(\mathcal{U}_f)$ **do**
7:             $\mathcal{H}_\mathcal{G} \leftarrow BERT\_Compose\_Sort(G, \zeta)$                  ▷ sort values within each group
8:   $\widehat{\mathcal{H}_\mathcal{G}} \leftarrow heading\_words(\mathcal{H}_\mathcal{G})$           ▷ replace sorted values of each group with their header group
9: **return** $BERT\_Compose\_Sort(\widehat{\mathcal{H}_\mathcal{G}}, \zeta)$                         ▷ sort heading groups
10:
11: **procedure** BERT_COMPOSE_SORT($\mathcal{A}, \zeta$)
    ▷ Sort input array of $\mathcal{A}$ through MLM
12:    **if** $||\mathcal{A}|| > \zeta$ **then**
13:       $\mathcal{C} \leftarrow Combine(\mathcal{A}, \zeta)$                    ▷ initial combination case generator with length of $\zeta$
14:       $\mathcal{I} \leftarrow BERT\_Base\_Sort(\mathcal{C})$                        ▷ sort initiated sequences
15:       $\mathcal{S} \leftarrow \mathcal{I}$                                    ▷ sort initiated sequences
16:       **for** $\mathcal{E}$ in $\mathcal{A} - \mathcal{I}$ **do**                       ▷ sequential adding the rest of elements
17:          $\mathcal{C} \leftarrow Seq(\mathcal{S}, \mathcal{E})$     ▷ generate new sequential cases by adding $\mathcal{E}$ in all $p$ positions of $\mathcal{S}$ ($p = [1, ..., ||\mathcal{S}|| + 1]$)
18:          $\mathcal{S} \leftarrow BERT\_Base\_Sort(\mathcal{C})$                  ▷ sort sequential cases
19:    **else**
20:       $\mathcal{C} \leftarrow Permute(\mathcal{A}, \zeta)$                  ▷ generate all new cases with all permutations of $\mathcal{A}$
21:       $\mathcal{S} \leftarrow BERT\_Base\_Sort(\mathcal{C})$               ▷ sort all possible permutation of $\mathcal{A}$
22: **return** $\mathcal{S}$
23:
24: **procedure** BERT_BASE_SORT($\mathcal{C}$)
    ▷ Sort input cases of $\mathcal{C}$ through MLM
25: Init MLM(Model)                            ▷ initialize a MLM model (i.e., Model="RoBERTa")
26:    **for** $\mathcal{C}_i$ in $\mathcal{C}$ **do**                                 ▷ process each case
27:       **for** $j$ in range($|\mathcal{C}_i|$) **do**
28:          $\mathcal{M}_{i,j} \leftarrow Mask(\mathcal{C}_{i,j})$                     ▷ generate mask pattern
29:          $\Theta_i \leftarrow \Theta_i + [\widehat{\mathcal{P}_i}(\mathcal{M}_{i,j}|S_i, \eta)]$       ▷ collect probability of each unmasked pattern
30:       $\Theta_i = \sum(\widehat{\mathcal{P}_i})$                               ▷ calculate score of each case
31:       **if** $\Theta_i > \Theta_m$ **then**:
32:          $\mathcal{C}_m = \mathcal{C}_i$
33:          $\Theta_m = \Theta_i$                            ▷ keep the best case with highest score
      **return** $\mathcal{C}_m$

---

Table 1: Evaluation of ordinal value detection on 212 features of 10 data sets

| | | Predicted | | |
|---|---|---|---|---|
| | | Positive | Negative | Total |
| Actual | Positive | 42 | 0 | 42 |
| | Negative | 6 | 164 | 170 |
| | Total | 48 | 164 | 212 |

## 4.1 Evaluation of Detecting Ordinal Features

We define *Ordinal Value Detection* function to apply BERT-Sort effectively, where it checks whether a set of unique values of a given feature corresponds to a semantic order. We select the ordinal values with length of 3 to 20, and remove numerical values in string format as part of pre-processing stage. Then, we shuffle input unique values, and generate $m$ number of sample cases. If $Avg(\Theta_{1..m}) > 1e-4$, BERT-Sort consider that there is an ordinal relationship between values. In our experiment, we use benchmark that includes 10 data sets with 212 features. In this experiment, $m = 3$ and $\eta = 20,000$ in *BERT_Base_Sort()* and initiate MLM on *RoBERTa* model. As shown in Table 1, the ordinal value detection function predicts ordinal values with $Precision = 0.875$, $Recall = 1$ and $F1 = 0.933$. As shown in Table 1 and Table 2-row#5, BERT-Sort detects 6 false-positive features out of 212 features due to possibility of generating a probability score for a set of given values in zero-shot environment.

Table 2: Five examples of the ordinal value detection

| # | Data set | Unique Values | Ground Truth | Detection Status |
|---|----------|---------------|--------------|------------------|
| 1 | uci-audiology-original | ['Normal', 'Moderate', 'Mild'] | Yes | True-Positive |
| 2 | cat-in-the-dat-ii | ['Warm', 'Boiling Hot', 'Freezing', 'Lava Hot', 'Hot', 'Cold'] | Yes | True-Positive |
| 3 | uci-automobile | ['Wagon', 'Sedan', 'Hatchback', 'Hardtop', 'Convertible'] | No | True-Negative |
| 4 | uci_automobile | ['Blue Collar', 'Management', 'Entrepreneur', ..., 'Admin'] | No | True-Negative |
| 5 | cat-in-the-dat-ii | ['Circle', 'Polygon', 'Square', 'Star', 'Trapezoid', 'Triangle'] | No | **False-Positive** |

Table 3: A comparison between a classical accuracy metric ($Acc$) and an ordinal accuracy metric ($Ord_{Acc}$) for ground truth ordinal values of $[Jan \prec Feb \prec Mar \prec Apr]$

| # | Ranked Values | $Acc$ | $Ord_{Acc}$ |
|---|---------------|-------|-------------|
| 1 | $[\textbf{Feb} \prec \textbf{Jan} \prec Mar \prec Apr]$ | 0.5 | 0.87 |
| 2 | $[\textbf{Mar} \prec Feb \prec \textbf{Jan} \prec Apr]$ | 0.5 | 0.75 |

Even though BERT-Sort may encode those false-positive detected features, the encoded values may not cause negative impacts to down-stream machine learning tasks.

## 4.2 Evaluation of Encoding Ordinal Values

In addition to classical evaluation of classification problem, such as *Accuracy* (Mosley, 2013)(Urbanowicz and Moore, 2015), we proposed a new metric based on ordinal value error rate where it calculates the ordinal distance error between the ground truths and predictions as follows.

$$Ord_{Acc} = \sum_{i=1}^{\|A\|} \frac{\|A\| - |\mathcal{L}_i - \widehat{\mathcal{L}}_i|}{\|A\|^2} \qquad (2)$$

Let $\|\mathcal{A}\|$ denotes the length of input array (unique values of a given feature, $\mathcal{F}$), $|\mathcal{L}_i - \widehat{\mathcal{L}}_i|$ is the absolute distance of $i$-th element in predicted order ($\widehat{\mathcal{L}}_i$) from the actual position ($\mathcal{L}_i$) in ground truth data set, and $Ord_{Acc}$ represents the *Ordinal Accuracy*. For instance, as shown in Table 3, row#1, both values of *Jan* and *Feb* have distance (error rate) of one to their actual positions and thus $Ord_{Acc} = 0.87$. On the other hand, the accuracy for this example is 0.5 because two of four elements are incorrect. In contrast, another example shown in row #2 also has the same accuracy equal to 0.5. However, the second example has worse *Ordinal Accuracy* where $Ord_{Acc} = 0.75$ in compared to the first one. As shown row #2 has a longer distance (higher error rate) for both *Jan* and *Mar* to their ground truth positions. Although we evaluate the benchmark on both evaluation metrics, we recommend $Ord_{Acc}$ for ordinal evaluation.

Table 4 shows the evaluation results on 10 different data sets for 42 distinctive features of annotated ground truth of ordinal values where $\zeta = 4$, in our benchmark $Avg(\|\mathcal{A}\|) = 4$. $Ord_{Acc}$ corresponds to ordinal accuracy and $Acc$ corresponds to classical accuracy metric. As part of the pre-processing stage, we consider a set of special categories where it can be removed from unique values due to general information or non-ordinal values, such as: "?", Null string, Null value, "unknown", "unmeasured", etc. We remove special categories from both BERT-Sort and *OrdinalEncoder* to have an unbiased comparison. As shown in this table, BERT-Sort improves the performance in terms of both $Ord_{Acc}$ and $Acc$ on all four different initiated MLMs. BERT-Sort with $M_2$ initialization (RoBERTa) achieves the best performance with significant improvements of 27% and 55% against the baseline based on $Ord_{Acc}$ and $Acc$ metrics, respectively.

## 4.3 AutoML Evaluation.

We use 4 AutoML platforms including AutoGluon, FLAML, H2O, and MLJAR to evaluate the effectiveness of encoded categorical ordinal features by using BERT-Sort to the ultimate machine learning performance. We evaluate 5 different versions of 10 data sets, where each method encodes

Table 4: A Comparisons of semantic ordinal value evaluation of BERT-Sort with initiation on 4 different MLMs of DistilBERT, RoBERTa, XLM, BERT-base-uncased ($M_{1..4}$, respectively), and OrdinalEncoder with two metrics of Ordinal Accuracy ($Ord_{Acc}$) and classical Accuracy ($Acc$) metrics on 10 different data sets and 42 distinctive features; (champions marked in **bold**; 🏆 indicates BERT-Sort feature champion in all models based on both metrics; 🏆 indicates OrdinalEncoder feature champion against BERT-Sort based on both metrics)

| Evaluation | $Ord_{Acc}$ | | | | | $Acc$ | | | | |
|---|---|---|---|---|---|---|---|---|---|---|
| Approach | BERT-Sort | | | | OrdinalEncoder | BERT-Sort | | | | OrdinalEncoder |
| Model | $M_1$ | $M_2$ | $M_3$ | $M_4$ | | $M_1$ | $M_2$ | $M_3$ | $M_4$ | |
| Feature | | | | | | | | | | |
| $\mathcal{F}_1$ | **0.92** | **1.00** | 0.76 | **0.84** | 0.76 | **0.60** | **1.00** | 0.40 | 0.40 | 0.00 |
| $\mathcal{F}_2$ 🏆 | **1.00** | **1.00** | **1.00** | **1.00** | 0.50 | **1.00** | **1.00** | **1.00** | **1.00** | 0.00 |
| $\mathcal{F}_3$ 🏆 | **1.00** | **1.00** | **1.00** | **1.00** | 0.50 | **1.00** | **1.00** | **1.00** | **1.00** | 0.00 |
| $\mathcal{F}_4$ | **1.00** | **1.00** | 0.78 | 0.78 | 0.56 | **1.00** | **1.00** | 0.33 | 0.33 | 0.00 |
| $\mathcal{F}_5$ 🏆 | **1.00** | **1.00** | **1.00** | **1.00** | 0.50 | **1.00** | **1.00** | **1.00** | **1.00** | 0.00 |
| $\mathcal{F}_6$ 🏆 | **1.00** | **1.00** | **1.00** | **1.00** | 0.50 | **1.00** | **1.00** | **1.00** | **1.00** | 0.00 |
| $\mathcal{F}_7$ 🏆 | **1.00** | **1.00** | **1.00** | **1.00** | 0.50 | **1.00** | **1.00** | **1.00** | **1.00** | 0.00 |
| $\mathcal{F}_8$ | 0.84 | **1.00** | **1.00** | 0.84 | 0.76 | 0.40 | **1.00** | **1.00** | 0.60 | 0.20 |
| $\mathcal{F}_9$ | 0.76 | 0.76 | 0.76 | **0.84** | 0.68 | 0.00 | 0.00 | 0.20 | **0.40** | **0.20** |
| $\mathcal{F}_{10}$ | **0.94** | 0.78 | 0.78 | 0.89 | 0.72 | **0.67** | 0.00 | 0.17 | **0.67** | 0.33 |
| $\mathcal{F}_{11}$ | 0.78 | 0.78 | 0.78 | **1.00** | 0.56 | 0.33 | 0.33 | 0.33 | **1.00** | 0.33 |
| $\mathcal{F}_{12}$ | 1.00 | 1.00 | 0.78 | 0.78 | 1.00 | 1.00 | 1.00 | 0.33 | 0.33 | 1.00 |
| $\mathcal{F}_{13}$ | 0.74 | **0.93** | 0.72 | **0.82** | 0.72 | **1.00** | 0.50 | 0.00 | 0.25 | 0.00 |
| $\mathcal{F}_{14}$ | 0.75 | **1.00** | **1.00** | 0.88 | 0.75 | 0.00 | **1.00** | **1.00** | 0.50 | 0.25 |
| $\mathcal{F}_{15}$ 🏆 | **1.00** | **1.00** | **1.00** | 0.88 | 0.75 | **1.00** | **1.00** | **1.00** | 0.50 | 0.25 |
| $\mathcal{F}_{16}$ | 1.00 | 1.00 | 1.00 | 1.00 | 1.00 | 1.00 | 1.00 | 1.00 | 1.00 | 1.00 |
| $\mathcal{F}_{17}$ 🏆 | 1.00 | 0.78 | 1.00 | 1.00 | **1.00** | 1.00 | 0.33 | 1.00 | 1.00 | **1.00** |
| $\mathcal{F}_{18}$ 🏆 | **1.00** | **1.00** | **1.00** | 0.78 | 0.56 | **1.00** | **1.00** | **1.00** | 0.33 | 0.33 |
| $\mathcal{F}_{19}$ 🏆 | 0.78 | **1.00** | **1.00** | 0.78 | 0.56 | 0.33 | **1.00** | **1.00** | 0.33 | 0.00 |
| $\mathcal{F}_{20}$ | 0.75 | **1.00** | **1.00** | 0.88 | 0.75 | 0.00 | **1.00** | **1.00** | 0.50 | 0.25 |
| $\mathcal{F}_{21}$ | 0.62 | 0.88 | **1.00** | 0.75 | 0.75 | 0.00 | 0.50 | **1.00** | 0.50 | 0.25 |
| $\mathcal{F}_{22}$ 🏆 | **1.00** | **1.00** | **1.00** | **1.00** | 0.56 | **1.00** | **1.00** | **1.00** | **1.00** | 0.33 |
| $\mathcal{F}_{23}$ 🏆 | 0.78 | **1.00** | **1.00** | 0.78 | 0.56 | 0.33 | **1.00** | **1.00** | 0.33 | 0.00 |
| $\mathcal{F}_{24}$ | **1.00** | **1.00** | **1.00** | **1.00** | 0.50 | **1.00** | **1.00** | **1.00** | **1.00** | 0.00 |
| $\mathcal{F}_{25}$ | 0.84 | **1.00** | **1.00** | 0.67 | 0.63 | 0.14 | **1.00** | **1.00** | 0.14 | 0.14 |
| $\mathcal{F}_{26}$ | 1.00 | 1.00 | 1.00 | 1.00 | 1.00 | 1.00 | 1.00 | 1.00 | 1.00 | 1.00 |
| $\mathcal{F}_{27}$ | 0.78 | **1.00** | **1.00** | **1.00** | 0.78 | 0.33 | **1.00** | **1.00** | **1.00** | 0.33 |
| $\mathcal{F}_{28}$ | **0.78** | 0.78 | 0.78 | **1.00** | 0.56 | 0.33 | 0.33 | 0.33 | **1.00** | 0.33 |
| $\mathcal{F}_{29}$ | 0.76 | **1.00** | 0.76 | 0.84 | 0.68 | 0.20 | **1.00** | 0.20 | **0.40** | 0.20 |
| $\mathcal{F}_{30}$ | 0.75 | **1.00** | 0.75 | 0.75 | 0.50 | **0.25** | **1.00** | 0.50 | 0.00 | 0.00 |
| $\mathcal{F}_{31}$ | 0.78 | **1.00** | **1.00** | 0.78 | 0.78 | 0.33 | **1.00** | **1.00** | 0.33 | 0.33 |
| $\mathcal{F}_{32}$ | 0.78 | 0.78 | 0.78 | 0.78 | 0.78 | 0.33 | 0.33 | 0.33 | 0.33 | 0.33 |
| $\mathcal{F}_{33}$ 🏆 | 0.76 | 0.68 | 0.84 | 0.84 | **0.92** | 0.00 | 0.00 | 0.60 | 0.60 | **0.60** |
| $\mathcal{F}_{34}$ 🏆 | **1.00** | **1.00** | **1.00** | 0.78 | 0.56 | **1.00** | **1.00** | **1.00** | 0.33 | 0.00 |
| $\mathcal{F}_{35}$ 🏆 | 0.78 | **1.00** | **1.00** | 0.78 | 0.56 | 0.33 | **1.00** | **1.00** | 0.33 | 0.00 |
| $\mathcal{F}_{36}$ | 0.78 | **1.00** | **1.00** | 0.78 | 0.56 | 0.33 | **1.00** | **1.00** | 0.33 | 0.00 |
| $\mathcal{F}_{37}$ | **1.00** | **1.00** | 0.78 | **1.00** | 0.78 | **1.00** | **1.00** | 0.33 | **1.00** | 0.33 |
| $\mathcal{F}_{38}$ | **1.00** | **1.00** | 0.78 | 0.78 | 0.78 | **1.00** | **1.00** | 0.33 | 0.33 | 0.33 |
| $\mathcal{F}_{39}$ | 0.75 | 0.75 | 0.88 | **0.88** | **0.88** | 0.25 | 0.00 | 0.50 | 0.50 | **0.50** |
| $\mathcal{F}_{40}$ | 0.78 | **1.00** | **1.00** | 0.78 | 0.56 | 0.33 | **1.00** | **1.00** | 0.33 | 0.33 |
| $\mathcal{F}_{41}$ | 0.78 | **1.00** | **1.00** | 0.78 | 0.56 | 0.33 | **1.00** | **1.00** | 0.33 | 0.33 |
| $\mathcal{F}_{42}$ 🏆 | **1.00** | **1.00** | **1.00** | **1.00** | 0.56 | **1.00** | **1.00** | **1.00** | **1.00** | 0.33 |
| **#champions** | 30 | **35** | 31 | 32 | 3 | 23 | **31** | 27 | 28 | 5 |
| | w.r.t. OrdinalEncoder | | | | w.r.t. $M_2$ | w.r.t. OrdinalEncoder | | | | w.r.t. $M_2$ |
| **Improvement** | 0.20 | **0.27** | 0.25 | 0.20 | baseline | 0.31 | **0.55** | 0.49 | 0.34 | baseline |

Table 5: Overall average F1 score and average Accuracy score performance of 8 original data sets, and its 4 other methods of ordinal value encoders on 4 AutoML platforms with 4 different randomization experiences (4 seeds)

| Method | F1 Score | Accuracy Score |
|---|---|---|
| Encoded BERT | 0.520 | 0.728 |
| OrdinalEncoder | 0.615 | 0.764 |
| Original | 0.625 | 0.769 |
| BERT-Sort | 0.636 | 0.784 |
| Human Annotation | 0.637 | 0.785 |

only *ordinal features* and leave the rest of features as-is. The following shows 5 different methods which transform the original data set to different encoded features as inputs to 4 AutoML platforms.
**Original**. It refers to the original data set without any changes (our baseline).
**Encoded BERT**. This method encodes the ordinal value by utilizing Sentence-BERT (Reimers and Gurevych, 2019) to generate a high-dimension continuous vector representation with a size 238. Then, we apply Scikit-Learn PCA (Szlam et al., 2014), a linear dimensionality reduction method based on Singular Value Decomposition(Maćkiewicz and Ratajczak, 1993), to transform the high-dimension vector into a low-dimension (a single value) that represents the ordinal feature.
**Human Annotation**. We manually assign the orders (i.e. integers) to ordinal values based on their semantic meanings. The annotated values are considered as ground-truth.
**OrdinalEncoder**. where it uses Scikit-Learn OrdinalEncoder to encode the ordinal values.
**BERT-Sort**. which encodes ordinal value based on BERT-Sort approach.

We split each input data set (each version of data set) into 75% which is sent to each AutoML platform, and 25% which is used to test the trained model. The experiments have been completed with 4 different seeds of *[108, 180, 234, 309]* to split data sets for training and testing (see Section 9 for detail). We limit the execution time of AutoML platforms to 5 minutes. Note that the data sets in our evaluation include all different features including numerical, text, etc. In this evaluation, we use F1 metric and accuracy evaluation. F1 metric has more restriction in compared to accuracy metric because it aggregates both *Recall* and *Precision* by considering the concept of harmonic mean (Grandini et al., 2020) (Takahashi et al., 2021). We use F1-macro where it is computes the average of F1 score of each class with weighting depends on the average parameter. All AutoML platforms failed on two data sets (regressions task) of *UCI_Coil_1999_Competition* where it requires 7 features predictions (7 of Algae frequency distributions must be determined), and *UCI-Automobile* which required additional pre-processing step. Although AutoGluon and H2O generate at least a model for these data sets but returns a negative score value which indicates that the model fits data poorly. We decided to remove these two data set to avoid bias evaluation across different AutoMLs. All AutoML platforms successfully generated at least a model (success) for data sets with classification task except H2O where it is failed on two data sets of *UCI Bank* and *Kaggle Cat-in-the-Dat-ii* data sets due to time limitation. Table 5 shows the overall performance of all data sets across 4 AutoML platforms. The performance results in this table indicates that *human annotation of ordinal values have the best performance* and **BERT-Sort was able to produce results with *F1 score* and *Accuracy score* close to manually annotated version of data sets** that has highest performance. In addition, manually transformed ordinal values (human annotated) shows that a correct encoding can improve ultimate end-to-end performance of machine-learning models. Table 6 shows the details of our experiments per AutoML platform. Although nominal values and other features (such as text, and numerical) may contributed to model prediction, BERT-Sort improves the performance of 3 AutoML platforms. The fine-grained evaluation of this table per AutoML platform per Data Set per Encoded Method is shown in Table 11 and Table 12.

Table 6: A comparison evaluation between different AutoML platforms on 8 benchmark data sets by using 5 methods of encoding for different experiments with 4 different random seeds

| AutoML | Method | F1 Score | Accuracy Score |
|---|---|---|---|
| AutoGluon | Encoded BERT | 0.560 | 0.766 |
| | OrdinalEncoder | 0.632 | 0.790 |
| | Original | 0.648 | 0.799 |
| | BERT-Sort | 0.640 | 0.788 |
| | Human Annotation | 0.614 | 0.774 |
| FLAML | Encoded BERT | 0.480 | 0.727 |
| | OrdinalEncoder | 0.632 | 0.772 |
| | Original | 0.598 | 0.742 |
| | BERT-Sort | 0.631 | 0.784 |
| | Human Annotation | 0.648 | 0.787 |
| H2O | Encoded BERT | 0.570 | 0.714 |
| | OrdinalEncoder | 0.613 | 0.744 |
| | Original | 0.666 | 0.768 |
| | BERT-Sort | 0.679 | 0.780 |
| | Human Annotation | 0.679 | 0.802 |
| MLJAR | Encoded BERT | 0.480 | 0.702 |
| | OrdinalEncoder | 0.582 | 0.746 |
| | Original | 0.599 | 0.768 |
| | BERT-Sort | 0.606 | 0.782 |
| | Human Annotation | 0.617 | 0.780 |

## 5 Discussions

As shown in Table 4, encoding ordinal values by BERT-Sort with initialization of any MLM indicates BERT-Sort is able to achieve the state-of-the-art semantic encoding performance on categorical features. On the other hand, these results show that widely used categorical encoding functions such as *OrdinalEncoder* will *lead to diminish the semantic order of ordinal values*.

Interestingly, BERT-Sort can go beyond expectation by sorting categorical data where it is too complex for data scientists to rank elements manually. For instance, by initiating BERT-Sort on *BioClinical BERT*(Alsentzer et al., 2019) it can sort the severity of cancer as *[Melanoma > Leukemia > Cancer]* without any supervision (a supervised approach used by Lausser et al. (2020) to generate this order). More details about the broader impact of BERT-Sort usages across different domains, languages (i.e., Chinese, Japanese and Spanish) and semantic image sorting are explained in Section 12. Note that evaluated data sets contains many other non-ordinal features including nominal, text, etc. Therefore, the evaluation results reflects only the generalization of BERT-Sort.

## 6 Conclusion

In this paper, we introduced BERT-Sort, a novel automated approach to detect and encode categorical ordinal features.We introduced a benchmark that includes 10 different public data sets with 42 different ordinal features. We conducted an extensive evaluation on the benchmark where BERT-Sort is initialized on four popular MLMs of DistilBERT, RoBERTa, XLM and BERT-base. BERT-Sort significantly improves the performance of the state-of-the-art categorical encoders (i.e., Scikit-Learn OrdinalEncoder) by 22%. We also evaluated the effectiveness of the encoded features by BERT-Sort on 4 AutoML platforms. We compared the performances of encoded data sets via BERT-Sort against 4 different versions the original data sets. The evaluation results show that the trained model based on BERT-Sort encoder achieved a performance close to a hard encoded feature by a data scientist.

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

## 7 Appendix A: Experiment Setup

**Configuration Environment**. Our BERT-Sort encoder process is completed on a machine with Ubuntu, an Intel(R) Xeon(R) Gold 5120 CPU @ 2.20GHz (56 cores) with 128 GB RAM and 2 TB disk. AutoML experiments are completed on a machine with 4 Intel(R) Xeon(R) Silver 4114 CPU @ 2.20GHz (total 40 cores) with 512 GB RAM and 2 TB disk. All experiments are developed in Python with version '3.8'. About 17 hours CPU computation is required to complete all experiments except reported results in Figure 9 that required 50 hours.

**Benchmark Data Sets**. We use 10 publicly available data sets with 42 distinct features as listed in Table 7. We select these data sets because i) each data set has at least a categorical ordinal value; ii) data sets are publicly available.

**Artifacts**. In addition to provided link to the resources in Table 7, we provide the following artifacts that allow researchers and practitioners to reproduce the results. i) a copy of raw data sets; ii ) annotated categorical features (Human Annotation), which is used to evaluate BERT-Sort as the results shown in Table 4; and iii) encoded raw data sets through BERT-Sort with each model ($M_{1..4}$). The artifacts can be found at: https://github.com/marscod/BERT-Sort/blob/main/README.md.

In the repository, each folder in *outputs/MLM* contains a configuration file as *'config.json'* with a set of keys of *['model', 'mask', 'separator', 'eta', 'lower', 'target_files', 'ground_truth', 'default_grouping', 'default_zeta', 'preprocess']*. The key of *'target_files'* represent task information such as data set filename, a URL reference, type of task (classification or regression for AutoML evaluation), type of evaluation metric (F1 or RMSE). The key of *'ground_truth'* is a dictionary where the keys are representing the feature name (if any) or feature index, and the values are a list of ranked ordinal values. As explained in Section 4.2, the values (such as "?", and null values) are appended in the beginning of the list, and null values have been ignored for evaluation process.

## 8 Appendix B: BERT-Sort and MLMs hyper-parameters

We use four publicly available MLMs to evaluate the benchmark data sets based on their ranked ordinal values as listed in Table 8. Since the majority of models outperformed in compared to OrdinalEncoder, we may use DistilBERT which is 60% faster (Sanh et al., 2019). In addition, the inference on MLM can be optimized on CPU to reach 2ms per case (Philipp Schmid, 2022).

In addition, we initiate BERT-Sort on diverse MLMs to demonstrate the broader impact of the proposed approach across different domains and languages (see Appendix E - Section 12). The list of MLMs and the parameters are listed in Table 9.

## 9 Appendix C: Evaluation Results Detail

### 9.1 Ordinal Encoding Evaluation

Figure 3 shows an example score computation for a given unique array of $\mathcal{A}_1 = [A, B, C]$. This figure shows the average computation score for the first case with $\mathcal{A} = 0.4$.

Figure 7 shows a heat map plot of 4 initialized MLM for evaluation results of Table 4. In this figure, X-axis represents (from left to right) $Ord_{Acc}$ of BERT-Sort, $Ord_{Acc}$ of OrdinalEncoder, Differences of $Ord_{Acc}$ metric, $Acc$ metric of BERT-Sort, $Acc$ metric of OrdinalEncoder, and Differences of $Acc$ metric, respectively. Y-axis represents the data set name:feature name or feature index.

### 9.2 AutoML Evaluations

First, we use Scikit-Learn *train_test_split* function[1] to split the given input data set of the benchmark into 75%/25%. We use the following random seeds in all experiences: *[108, 180, 234, 309]*. Second, as explained in Section 4, we use 4 different AutoML platforms which are publicly available to train a model on training data sets and evaluate each AutoML platform on test data set. The configuration and version of each tool is shown in Table 10.

---

[1]https://scikit-learn.org/stable/modules/generated/sklearn.model_selection.train_test_split.html

Table 7: Annotated 42 distinct features from 10 different public data sets for ordinal values evaluation

| $\mathcal{F}_i$ | Data set | Ordinal Feature (index/name) | $\|\|\mathcal{A}\|\|$ | Source |
|---|---|---|---|---|
| $\mathcal{F}_1$ | uci_audiology-original | 1 | 5 | UCI Link |
| $\mathcal{F}_2$ | uci_audiology-original | 3 | 4 | |
| $\mathcal{F}_3$ | uci_audiology-original | 4 | 4 | |
| $\mathcal{F}_4$ | uci_audiology-original | 5 | 5 | |
| $\mathcal{F}_5$ | uci_audiology-original | 7 | 3 | |
| $\mathcal{F}_6$ | uci_audiology-original | 58 | 4 | |
| $\mathcal{F}_7$ | uci_audiology-original | 59 | 4 | |
| $\mathcal{F}_8$ | uci_audiology-original | 63 | 7 | |
| $\mathcal{F}_9$ | kaggle_cat-in-the-dat-ii | $ord_1$ | 6 | Kaggle Link |
| $\mathcal{F}_{10}$ | kaggle_cat-in-the-dat-ii | $ord_2$ | 7 | |
| $\mathcal{F}_{11}$ | uci_bank_marketing (bank) | marital | 3 | UCI Link |
| $\mathcal{F}_{12}$ | uci_bank_marketing (bank) | education | 4 | |
| $\mathcal{F}_{13}$ | uci_bank_marketing (bank) | month | 12 | |
| $\mathcal{F}_{14}$ | uci_car_evaluation | vhigh | 4 | UCI link |
| $\mathcal{F}_{15}$ | uci_car_evaluation | vhigh.1 | 4 | |
| $\mathcal{F}_{16}$ | uci_car_evaluation | 2 | 4 | |
| $\mathcal{F}_{17}$ | uci_car_evaluation | 2.1 | 3 | |
| $\mathcal{F}_{18}$ | uci_car_evaluation | small | 3 | |
| $\mathcal{F}_{19}$ | uci_car_evaluation | low | 3 | |
| $\mathcal{F}_{20}$ | uci_car_evaluation | unacc | 4 | |
| $\mathcal{F}_{21}$ | uci_Coil_1999_Competition_Data | 0 | 4 | UCI link |
| $\mathcal{F}_{22}$ | uci_Coil_1999_Competition_Data | 1 | 3 | |
| $\mathcal{F}_{23}$ | uci_Coil_1999_Competition_Data | 2 | 3 | |
| $\mathcal{F}_{24}$ | uci_automobile | 5 | 3 | UCI link |
| $\mathcal{F}_{25}$ | uci_automobile | 15 | 7 | |
| $\mathcal{F}_{26}$ | uci_labor-relations | 0 | 4 | UCI link |
| $\mathcal{F}_{27}$ | uci_labor-relations | 11 | 4 | |
| $\mathcal{F}_{28}$ | uci_Nursery | 0 | 3 | UCI link |
| $\mathcal{F}_{29}$ | uci_Nursery | 1 | 5 | |
| $\mathcal{F}_{30}$ | uci_Nursery | 2 | 4 | |
| $\mathcal{F}_{31}$ | uci_Nursery | 4 | 3 | |
| $\mathcal{F}_{32}$ | uci_Nursery | 6 | 3 | |
| $\mathcal{F}_{33}$ | uci_Nursery | 8 | 5 | |
| $\mathcal{F}_{34}$ | uci_Post-Operative-Patient | 0 | 3 | UCI Link |
| $\mathcal{F}_{35}$ | uci_Post-Operative-Patient | 1 | 3 | |
| $\mathcal{F}_{36}$ | uci_Post-Operative-Patient | 3 | 3 | |
| $\mathcal{F}_{37}$ | uci_Post-Operative-Patient | 5 | 3 | |
| $\mathcal{F}_{38}$ | uci_Post-Operative-Patient | 6 | 3 | |
| $\mathcal{F}_{39}$ | uci_Pittsburgh_Bridges | 3 | 4 | UCI link |
| $\mathcal{F}_{40}$ | uci_Pittsburgh_Bridges | 5 | 4 | |
| $\mathcal{F}_{41}$ | uci_Pittsburgh_Bridges | 9 | 4 | |
| $\mathcal{F}_{42}$ | uci_Pittsburgh_Bridges | 10 | 4 | |
| | Average | | 4.186 | |

Table 8: The list of Masked Language Models (MLM) which are used for the ordinal value evaluation

| # | $M_i$ | Model | Mask | $\eta$ | Hugging Face Model (link) | Source |
|---|---|---|---|---|---|---|
| 1 | $M_1$ | DistilBERT | [MASK] | 20,000 | distilbert-base-uncased | (Sanh et al., 2019) |
| 2 | $M_2$ | RoBERTa | <mask> | 20,000 | roberta-large | (Liu et al., 2019a) |
| 3 | $M_3$ | XLM | <mask> | 20,000 | xlm-roberta-large | (Conneau et al., 2019) |
| 4 | $M_4$ | BERT-Base | [MASK] | 20,000 | bert-base-uncased | (Devlin et al., 2018) |

Table 9: Additional public Masked Language Models (MLM) to present broader impacts of BERT-Sort across different domains and languages

| # | $M_i$ | Model | Mask | $\eta$ | Hugging Face Model (link) | Source |
|---|---|---|---|---|---|---|
| 1 | $M_5$ | Bio_ClinicalBERT | ([MASK]) | 40,000 | emilyalsentzer/Bio_ClinicalBERT | (Alsentzer et al., 2019) |
| 2 | $M_6$ | ChineseBERT | [MASK] | 20,000 | hfl/chinese-bert-wwm-ext | (Liu et al., 2019a) |
| 2 | $M_7$ | JapaneseBERT | [MASK] | 20,000 | ALINEAR/albert-japanese-v2 | N/A |
| 4 | $M_8$ | SpanishBERT | [MASK] | 20,000 | dccuchile/bert-base-spanish-wwm-cased | (Cañete et al., 2020) |

| $\mathcal{C}_{1,1}$ | A | B | C | $\mathcal{S}_{1,1}^k$ | $\widehat{P}(.)$ | | | |
|---|---|---|---|---|---|---|---|---|
| $\mathcal{M}_{1,1}^1$ | 1 | 0 | 0 | [CLS][MASK],B,C.[EOS] | 0.2 | | | $\widehat{P}(\mathcal{C}_{i,j}^1\|S_{i,j}^1,\eta)$ |
| $\mathcal{M}_{1,1}^2$ | 0 | 1 | 0 | [CLS]A[MASK],C.[EOS] | | 0.6 | | $\widehat{P}(\mathcal{C}_{i,j}^2\|S_{i,j}^2,\eta)$ |
| $\mathcal{M}_{1,1}^3$ | 0 | 0 | 1 | [CLS]A,B,[MASK][EOS] | | | 0.4 | $\widehat{P}(\mathcal{C}_{i,j}^3\|S_{i,j}^3,\eta)$ |
| $\mathcal{C}_{1,2}$ | C | B | A | | | | | $\theta_1 = 0.4$ |
| ... | | ... | | | | | | $\theta_2 = 0.37$ |

Figure 3: An example of score computation for a single feature with 3 ordinal values of $\mathcal{A}_1 = [A, B, C]$

| # | AutoML | Version | Parameters |
|---|---|---|---|
| 1 | MLJAR[2] | 0.11.2 | *'total_time_limit':3600* |
| 2 | FLAML[3] | 1.0.0 | *'time_budget': 3600, 'metric': 'f1'\|'r2'* |
| 3 | TPOT[4] | 0.11.7 | *scoring='f1_macro'* |
| 4 | H2O AutoML[5] | 3.36.1.1 | *'max_models': 20, and 'max_runtime_secs': 3600* |
| 5 | AutoGluon[6] | 0.4.0 | *eval_metric:"f1"*(classification problem) *, presets:'best_quality', time_limit:3600* |

Table 10: AutoML Configurations

In addition to evaluation results in Table 6 (with 4 seeds and 5 minute time limitation), we conducted extensive experiences with 1, 4 and 5 different randomized seeds to split each data set with a time limitation of 30 minutes. The results are concluded in Figure 9. In overall, the average F1 performance of all AutoML platforms on raw data sets is 0.346 versus F1 score of **0.377** on encoded data sets via BERT-Sort.

Since there might be many features (i.e., numerical and text features beside ordinal features) affect the overall performance of an end-to-end scenario for evaluating the encoded data sets, we may use data sets that only rely on categorical features. As such example, we use *UCI Car Evaluation* data set where all features are encoded through i) OrdinalEncoder, ii) BERT-Sort to produce two versions of the original raw data set. Then, we apply 11 different machine-learning algorithms to both data sets. In this experiment, we use CatBoostClassifier from *CatBoost*[7] package with version '1.0.5' and other ML algorithms from *Scikit-Learn package* with version '1.0.0'. Figure 10 shows the results of this evaluation. This pure comparison shows that BERT-Sort encoder outperformed on **all algorithms** with an average F1 score of **0.897** in comparison to *OrdinalEncoder* with an average F1 score of 0.532.

### 9.3 Fine-grained AutoML Evaluation Results

Table 11 and Table 12 show the fine-grained evaluation results of Table 6 with F1 metric and Accuracy metric, respectively. Note that a comparison between values in this fine-grained evaluation results may not show a clear affect of semantic ordinal values since each experiment rely on many factors such as characteristics of data set (i.e., the number of features, importance of ordinal features), AutoML hyper-parameters, AutoML approach to train a model, etc.

## 10 Appendix D: MLM Input Structure

As discussed in Section 4, once a case ($\mathcal{C}_i$), a set of ranked ordinal values, and its mask pattern ($\mathcal{P}_{i,k}$) have been generated, BERT-Sort produces an input similar to a sentence structure where it consists of the ordinal value and masked element. The generated structure tokenized and submitted to the initialized MLM for unmasking process. BERT-Sort may generate different sentences based on different separators for the unique values of *[Jan, Feb, Mar]*, such as "," or blank space " " or combination of both (", ") to separate values. Similarly, "." can be added to the end of sentence or can

---

[7] http://catboost.ai/

Table 11: **F1 Score** of AutoML evaluations per AutoML platform per data set per encoded method

| AutoML | Data Set | F1 Score of Encoded Method | | | | |
|---|---|---|---|---|---|---|
| | | BERT-Sort | Embeded BERT | Human Annotation | Ordinal Encoder | Original |
| AutoGluon | Nursery | 1.000 | 0.974 | 1.000 | 1.000 | 1.000 |
| | Pittsburgh_Bridges | 0.511 | 0.384 | 0.418 | 0.495 | 0.535 |
| | Post-Operative | 0.154 | 0.131 | 0.154 | 0.154 | 0.134 |
| | audiology | 0.543 | 0.474 | 0.543 | 0.565 | 0.515 |
| | bank | 0.715 | 0.701 | 0.702 | 0.714 | 0.721 |
| | car_eval | 0.987 | 0.533 | 0.988 | 0.978 | 0.990 |
| | cat-in-the-dat-ii | 0.451 | 0.458 | 0.461 | 0.449 | 0.462 |
| | labor-relations | 0.762 | 0.828 | 0.647 | 0.699 | 0.828 |
| FLAML | Nursery | 0.949 | 0.695 | 0.949 | 0.950 | 0.900 |
| | Pittsburgh_Bridges | 0.415 | 0.220 | 0.393 | 0.419 | 0.295 |
| | Post-Operative | 0.302 | 0.198 | 0.386 | 0.361 | 0.272 |
| | audiology | 0.522 | 0.389 | 0.518 | 0.484 | 0.398 |
| | bank | 0.661 | 0.669 | 0.676 | 0.673 | 0.657 |
| | car_eval | 0.984 | 0.478 | 0.980 | 0.978 | 0.983 |
| | cat-in-the-dat-ii | 0.568 | 0.562 | 0.587 | 0.564 | 0.567 |
| | labor-relations | 0.646 | 0.629 | 0.697 | 0.625 | 0.715 |
| H2O | Nursery | 1.000 | 0.913 | 0.950 | 0.950 | 0.950 |
| | Pittsburgh_Bridges | 0.631 | 0.515 | 0.592 | 0.558 | 0.561 |
| | Post-Operative | 0.295 | 0.268 | 0.359 | 0.257 | 0.379 |
| | audiology | 0.568 | 0.445 | 0.530 | 0.503 | 0.560 |
| | car_eval | 0.971 | 0.583 | 0.989 | 0.969 | 0.966 |
| | labor-relations | 0.608 | 0.698 | 0.656 | 0.442 | 0.579 |
| MLJAR | Nursery | 0.770 | 0.715 | 0.902 | 0.913 | 0.913 |
| | Pittsburgh_Bridges | 0.439 | 0.233 | 0.454 | 0.279 | 0.411 |
| | Post-Operative | 0.266 | 0.238 | 0.336 | 0.223 | 0.236 |
| | audiology | 0.524 | 0.296 | 0.498 | 0.428 | 0.409 |
| | bank | 0.695 | 0.676 | 0.711 | 0.687 | 0.708 |
| | car_eval | 0.985 | 0.563 | 0.907 | 0.987 | 0.978 |
| | cat-in-the-dat-ii | 0.551 | 0.500 | 0.510 | 0.515 | 0.515 |
| | labor-relations | 0.621 | 0.621 | 0.621 | 0.621 | 0.621 |

Table 12: **Accuracy score** of AutoML evaluations per AutoML platform per data set per encoded method with 4 different 4 experiments (4 seeds)

| AutoML | Data Set | Accuracy of Encoded Method | | | | |
| | | BERT-Sort | Embeded BERT | Human Annotation | Ordinal Encoder | Original |
|---|---|---|---|---|---|---|
| AutoGluon | Nursery | 1.000 | 0.978 | 1.000 | 1.000 | 1.000 |
| | Pittsburgh_Bridges | 0.648 | 0.546 | 0.583 | 0.630 | 0.667 |
| | Post-Operative | 0.337 | 0.326 | 0.337 | 0.337 | 0.337 |
| | audiology | 0.815 | 0.790 | 0.815 | 0.850 | 0.805 |
| | bank | 0.899 | 0.900 | 0.899 | 0.901 | 0.897 |
| | car_eval | 0.995 | 0.900 | 0.995 | 0.992 | 0.995 |
| | cat-in-the-dat-ii | 0.813 | 0.813 | 0.814 | 0.813 | 0.814 |
| | labor-relations | 0.800 | 0.875 | 0.750 | 0.800 | 0.875 |
| FLAML | Nursery | 1.000 | 0.934 | 0.999 | 1.000 | 1.000 |
| | Pittsburgh_Bridges | 0.481 | 0.333 | 0.407 | 0.444 | 0.361 |
| | Post-Operative | 0.587 | 0.457 | 0.576 | 0.554 | 0.467 |
| | audiology | 0.790 | 0.755 | 0.820 | 0.790 | 0.620 |
| | bank | 0.897 | 0.900 | 0.899 | 0.896 | 0.895 |
| | car_eval | 0.997 | 0.917 | 0.996 | 0.995 | 0.992 |
| | cat-in-the-dat-ii | 0.822 | 0.823 | 0.824 | 0.824 | 0.822 |
| | labor-relations | 0.700 | 0.700 | 0.775 | 0.675 | 0.775 |
| H2O | Nursery | 1.000 | 0.922 | 1.000 | 1.000 | 1.000 |
| | Pittsburgh_Bridges | 0.676 | 0.602 | 0.648 | 0.639 | 0.639 |
| | Post-Operative | 0.565 | 0.511 | 0.685 | 0.554 | 0.554 |
| | audiology | 0.820 | 0.785 | 0.805 | 0.830 | 0.820 |
| | car_eval | 0.991 | 0.762 | 0.997 | 0.991 | 0.992 |
| | labor-relations | 0.625 | 0.700 | 0.675 | 0.450 | 0.600 |
| MLJAR | Nursery | 0.969 | 0.886 | 0.951 | 0.966 | 0.966 |
| | Pittsburgh_Bridges | 0.463 | 0.343 | 0.537 | 0.315 | 0.528 |
| | Post-Operative | 0.641 | 0.565 | 0.663 | 0.598 | 0.587 |
| | audiology | 0.775 | 0.535 | 0.735 | 0.695 | 0.670 |
| | bank | 0.892 | 0.890 | 0.884 | 0.887 | 0.888 |
| | car_eval | 0.997 | 0.883 | 0.960 | 0.996 | 0.992 |
| | cat-in-the-dat-ii | 0.816 | 0.811 | 0.809 | 0.810 | 0.810 |
| | labor-relations | 0.700 | 0.700 | 0.700 | 0.700 | 0.700 |

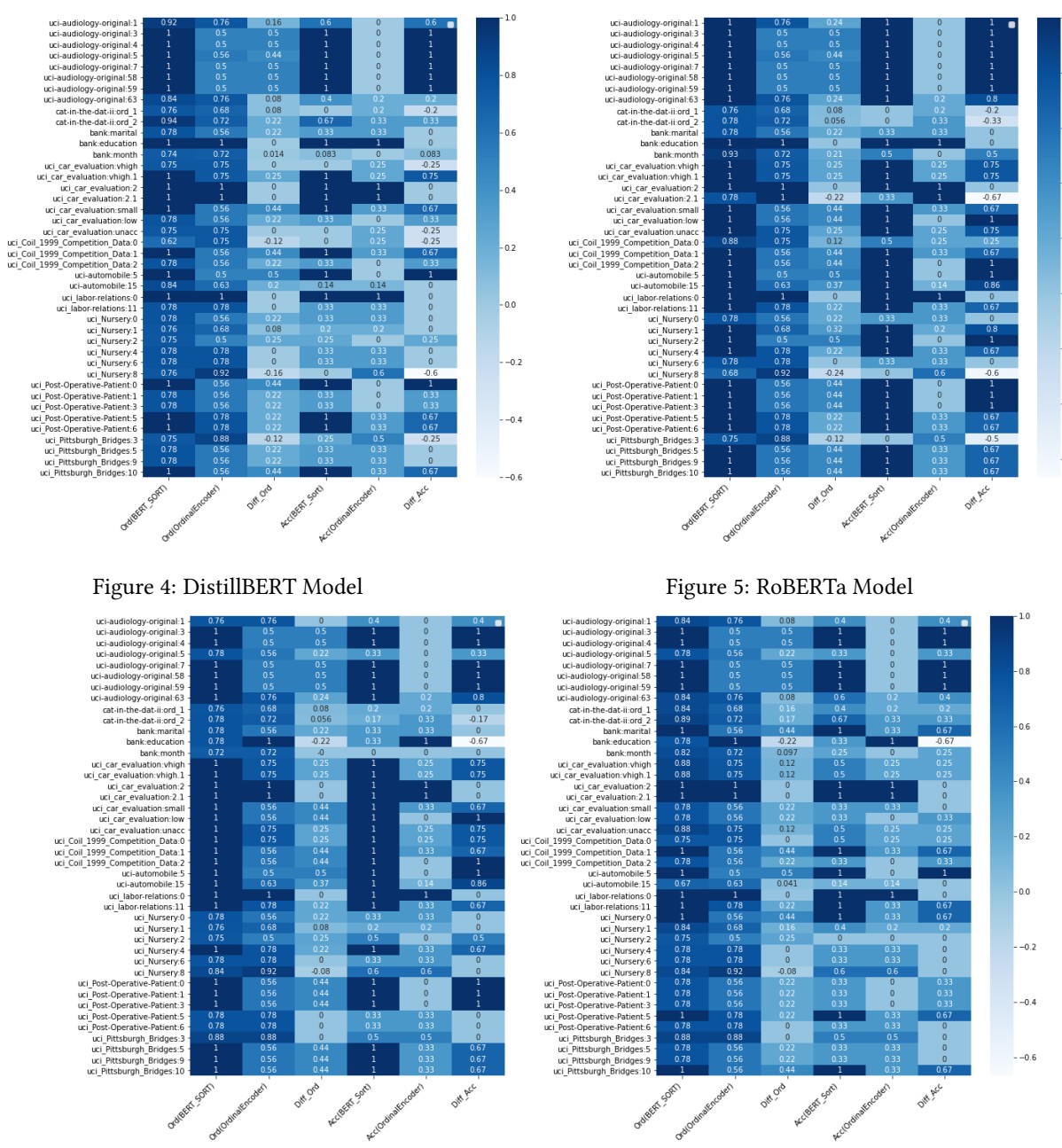

Figure 4: DistillBERT Model

Figure 5: RoBERTa Model

Figure 6: XLM

Figure 7: BERT-base Model

Figure 8: BERT-Sort results with different MLM initialization

be dropped. For example, BERT-Sort can generate *"Jan, [MASK], Mar."* for a separator of ", " or *"Jan [MASK] Mar."* for a separator of " "(blank space). We conduct an extensive empirical study to find the best pattern to construct different inputs to MLM. As shown previously (Table 4) BERT-Sort with initialization of RoBERTa is outperformed in compared to other MLMs. We construct different input structures only on RoBERTa. Table 13 shows the results of this empirical study where it shows the total number of champions based on $Ord_{Acc}$ metric and $Acc$ metric for 42 distinct features in our

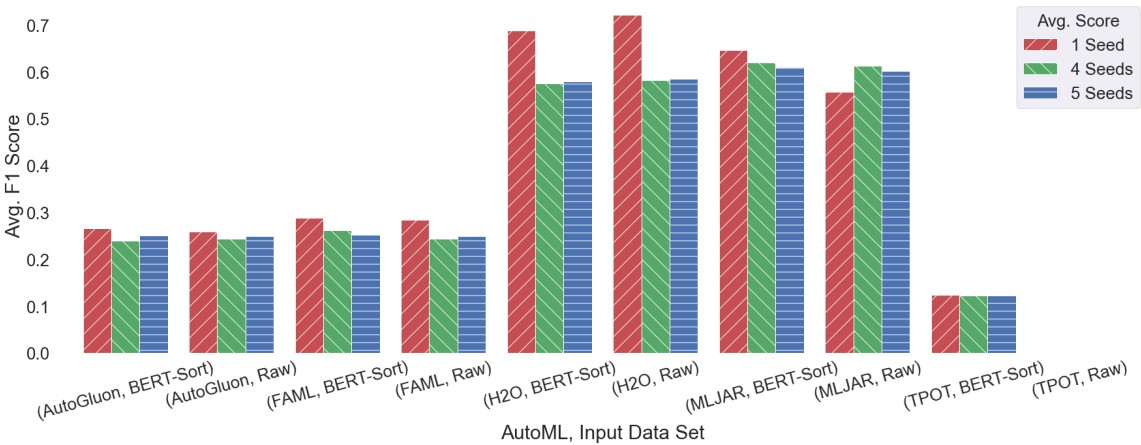

Figure 9: Performance of 5 AutoML tools on BERT-Sort-based encoded data sets and raw data sets with 1, 4 and 5 different seeds

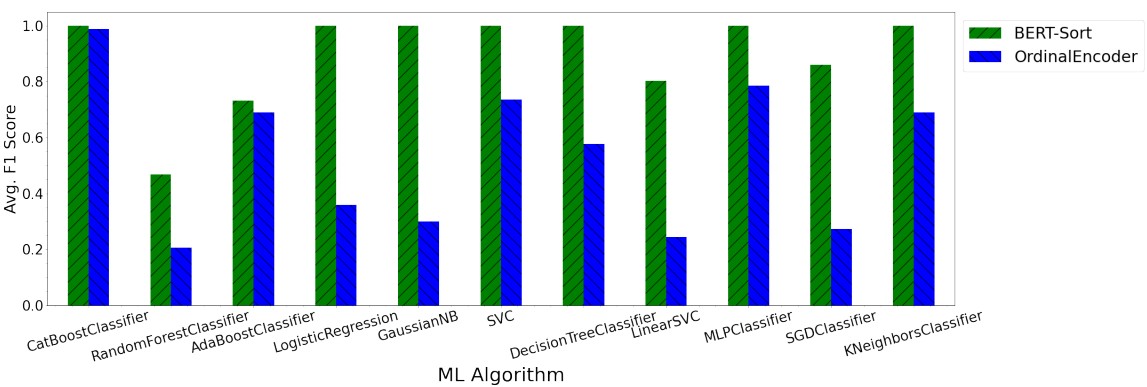

Figure 10: Performance of 11 ML algorithms on encoded the original *UCI Car Evaluation* data set through: i) BERT-Sort Encoder and ii) OrdinalEncoder

benchmark data sets. The results show that adding a comma between values and adding "." at the end construct best practice with $Ord_{Acc} = 0.27$ and $Acc = 0.55$ improvement over OrdinalEncoder.

Table 13: A comparison between BERT-Sort and OrdinalEncoder for sorting ordinal values with different input structures using RoBERTa-MLM

| # | Input Structure | | #Champions | | | | BERT-Sort Improvement | |
|---|---|---|---|---|---|---|---|---|
| | | | based on $Ord_{Acc}$ | | based on $Acc$ | | | |
| | Separator | End | BERT-Sort | OrdinalEncoder | BERT-Sort | OrdinalEncoder | $Ord_{Acc}$ | $Acc$ |
| 1 | ', ' (comma & blank space) | '.' | 34 | 3 | 30 | 5 | 0.26 | 0.51 |
| 2 | ' ' (blank space) | '.' | 32 | 3 | 26 | 5 | 0.20 | 0.35 |
| 3 | ',' (comma) | '.' | 35 | 3 | 31 | 5 | 0.27 | 0.55 |
| 4 | ', ' (comma & blank space) | '' (null string) | 32 | 2 | 25 | 4 | 0.24 | 0.46 |
| 5 | ' ' (blank space) | "" (null string) | 33 | 5 | 29 | 5 | 0.23 | 0.44 |
| 6 | ', ' (comma) | '' (null string) | 29 | 4 | 25 | 3 | 0.23 | 0.44 |

## 11 Appendix E: BERT-Sort Acceleration Parameters

In Section 3.2, we introduce two scaling approaches for handling BERT-Sort with large number of ordinal values for saving computation time. In an ideal case, the native BERT-Sort can generate the

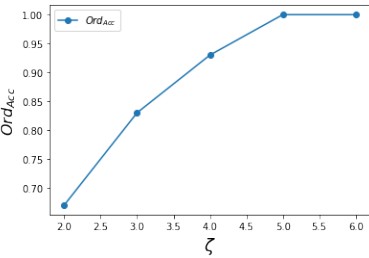

Figure 11: A comparison between BERT-Sort outputs on a feature with randomized values of 12 months abbreviations (['Jan', 'Feb', ..., 'Dec']) with $\zeta$ parameter in range of 2 to 5

best results without two scaling approaches because it finds the best ordinal values by checking all different possibilities. Although most of the ordinal features do not have more than 15 values (Table 7 indicates that the average number of ordinal values in our benchmark is 4), the native BERT-Sort has $\mathcal{O}(n!)$ computation which makes the algorithm impossible to be executed for a large number of ordinal values of a feature. The first approach of BERT-Sort acceleration is grouping where it can be easily disabled as a hyper-parameter. In the second acceleration, we recommend selecting maximum possible value for $\zeta$ (based on available resources, i.e. CPU cores) where it disables or postpone scaling algorithms as much as possible. As one example, Figure 11 shows BERT-Sort results on a feature with randomized values of 12 months abbreviations (['Jan', 'Feb', ..., 'Dec']) with $\zeta$ parameter in range of 2 to 5, where $\zeta \geq 5$ sorts all elements correctly.

Scaling algorithms may reduce the performance of BERT-Sort, but it helps to reduce computation time. An alternative approach for scaling BERT-Sort is parallel implementations because each case does not rely on other cases, and all BERT-Sort agents may generate the score of their cases at the same time.

## 12 Appendix F: Multilingual and Multi-domain Sorting

In this section, we demonstrate the broader impact of the zero-shot of BERT-Sort across different domains (e.g., medical), different languages of ordinal values (i.e., English, Spanish, Japanese and Chinese).

BERT-Sort approach finds the semantic orders of the ordinal values based the highest probability (top 1) of cases as explained in Section 3. Therefore, BERT-Sort easily can be extended by initializing the algorithm on different pre-trained MLMs to sort elements across different domains and languages. Table 14 showcases how easily the same algorithm applied toward multilingual and diverse domains. For instance, the first row compares the output of BERT-Sort against OrdinalEncoder with top score of BERT-Sort of 0.961205.

Different automated approaches can be used for switching between pre-trained MLMs in BERT-Sort. We may automate selecting pre-trained MLM process by using a domain or language classification (i.e., detecting English language vs Spanish) to select the best model for a specific set of ordinal values. We may also use the score to easily switch between MLMs in BERT-Sort. For instance, BERT-Sort which is initiated by English RoBERTa returns $\Theta_M = 0$ on a set of Spanish ordinal values or on a specific domain (e.g., $\Theta_M = 0$ in row#6 where it is using generic English MLM, RoBERTa). However, row#7 shows the same input on BioClinicalBERT(Alsentzer et al., 2019) where BERT-Sort ranks values based on cancer's severity correctly. The automated process of switching between languages can be applied toward different domains. For example, BERT-Sort returns $\Theta_M = 0$ score on a pre-trained English RoBERTa (Liu et al., 2019b) for input values of [*Leukemia, Cancer, Melanoma*] but after switching the model to BioClinicalBERT(Alsentzer et al., 2019) BERT-Sorts returns correctly sorted elements where $\Theta_M > 0$. Row#8 to 10 show the output of BERT-Sort for given input in Japanese, Spanish, and Chinese languages where the same algorithm

was able to sort all values accurately ($Ord_{Acc} = 1.0$) by initializing BERT-Sort on different MLMs. The red color text highlights misplaced ordinal values in OrdinalEncoder.

Furthermore, BERT-Sort can be applied in multi modal environment. As such example, a set of raw images can be annotated through CLIP (Radford et al., 2021) and the unique values of annotated images can be sorted semantically without user's annotation through BERT-Sort as shown in Figure 12. As another example, BERT-Sort is capable of sorting any categorical information such as number words (i.e., *["One","Two", "Four"]*), that might be captured from an OCR process.

Table 14: A comparisons between outputs of BERT-Sort (with different initialized MLMs), and OrdinalEncoder across different domains and languages

| # | Input | Model | BERT-Sort (top 1) vs OrdinalEncoder | $\Theta_M$ |
|---|---|---|---|---|
| 1 | [Mar, Jan, Feb, May] | RoBERTa-large (Liu et al., 2019b) | [Jan<Feb<Mar<May]::[Feb<Jan<Mar<May] | 0.961205 |
| 2 | [Lava Hot, Hot, Boiling Hot] | RoBERTa-large | [Hot < Boiling Hot < Lava Hot]::[Boiling Hot < Hot < Lava Hot] | 0.977791 |
| 3 | [Eight, Four, Two, Six, Twelve] | RoBERTa-large | [Two < Four < Six < Eight < Twelve]::[Eight < Four < Six < Twelve < Two] | 0.774939 |
| 4 | [Low, Medium, High] | RoBERTa-large | [Low < Medium < High] ::[High < Low < Medium] | 0.927987 |
| 5 | [Blue, Red, Green] | RoBERTa-large | [Red < Green < Blue] :: [Blue< Green < Red] | 0.742441 |
| 6 | [Leukemia, Cancer, Melanoma] | RoBERTa-large | N/A::<Cancer < Leukemia < Melanoma> | 0.0 |
| 7 | [Leukemia, Cancer, Melanoma] | BioClinical BERT(Alsentzer et al., 2019) | [Melanoma< Leukemia < Cancer]:: [Cancer < Leukemia < Melanoma] | 0.001098 |
| 8 | [優れた, 貧しい,良い ] | Japanese BERT-MLM[8] | [貧しい < 良い < 優れた] :: [優れた < 良い < 貧しい] | 0.000162 |
| 9 | [Muy Buena, Normal, Buena] | Spanish BERT-MLM(Canete et al., 2020) | [Normal < Buena < Muy Buena] ::[Buena < Muy Buena < Normal] | 0.000288 |
| 10 | [差, 好, 优秀] | Chinese BERT-WWM(Cui et al., 2019) | [优秀 < 好 < 差] :: [优秀 < 好 < 差 ] | 0.617564 |

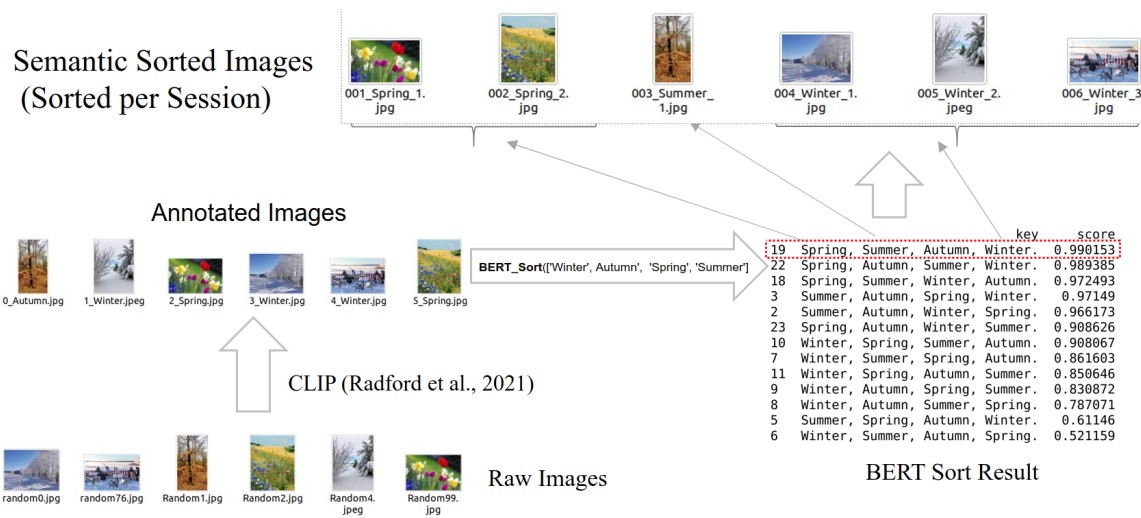

Figure 12: An example of an unsupervised image semantic sorting, where the images are labeled through CLIP (Radford et al., 2021) and ordinal values are sorted per sessions through BERT-Sort

## 13 Appendix G: AutoML Failures

**Assumptions**. Our evaluation is based on fitting 75% of data sets into different AutoML platforms and predicting 25% of the same test data set on all AutoMLs. Therefore, we did not consider any update on any specific AutoML platform beside using the same hyper-parameters such as *evaluation metric* and etc. The same configuration allows us to have a fair comparison between different AutoML platforms according to given input data sets. Therefore, we assume that any small fixes beside system configuration are out of scope in our evaluation. In addition, AutoML platforms are evaluated based on our split test data set which has been used to test all AutoML platforms. We

also use prediction function of each AutoML platform that possibly handle missing data, unknown values and etc. We list the most frequent exception errors as follows.

We observed several issues where an AutoML platform was not able to produce any model (failure cases). We explain the detail of errors for each AutoML platform as follows.

## 13.1 AutoGluon

AutoGluon failed to generate a model when it raises error of "`ValueError('AutoGluon did not successfully train any models')`". The following shows different failure reasons to produce a model.

`ValueError: Target is multiclass but average='binary'`. AutoGluon could not set parameters automatically for multi-class classification and binary classification. It is required to be set based on given input data set. This error has been raise in data sets *Audiology* (seed=180), *Pittsburgh_Bridges* (seed=180) data sets. `ValueError: cannot reshape array of size X into shape (Y,newaxis)` AutoGluon failed to generate model when it is rely on 'fold_fitting_strategy.py'. This issue raised in "audiology (seed:108)" `raise value.as_instanceof_cause() -> ray.exceptions.RayTaskError(ValueError)` This is a known issue in Dask-on-Ray[9] despite we are using the latest version of Dask. `RuntimeError: CUDA error: device-side assert triggered CUDA kernel errors might be asynchronously reported at some other API call,so the stacktrace below might be incorrect`. It is an issue when there is inconsistency between the total number of outputs and the total number of classes. Observed this issue in *cat-in-the-dat-ii*, *bank*, *car_eval*, *uci-automobile* data sets.

## 13.2 FLAML

`The max_iter was reached which means the coef_ did not converge` Training data normalization is required.

## 13.3 H2O

`failed: java.lang.ArrayIndexOutOfBoundsException: Index 64 out of bounds for length 6` Training data normalization is required.

## 13.4 MLJAR

`failed: Skip mix_encoding because no parameters were generated`. It is raised when there is missing load of already trained models after training restore (a known issued[10], despite using the latest version.

---

[9]https://github.com/ray-project/ray/issues/10124
[10]https://github.com/mljar/mljar-supervised/issues/185

