# OpenReview forum: "BERT-Sort: A Zero-shot MLM Semantic Encoder on Ordinal Features for AutoML"
_automl.cc/AutoML/2022/Track/Main — AutoML-Conf 2022 (Main Track)_

### Official Review · Reviewer_96Ap · 2022-03-15

**Potential Impact On The Field Of Automl:** 1. Being the first to provide an auto…
**Potential Impact On The Field Of Automl Rating:** 3
**Technical Quality And Correctness:** The experimental details seem correct.
**Technical Quality And Correctness Rating:** 3
**Clarity:** The paper is well written.
**Clarity Rating:** 3

**Summary Of Contributions:**

The authors introduce the BERT-Sort algorithm to semantically encode ordinal categorical values via a zero-shot pre-trained masked language model. The authors also construct benchmarks from 10 public sets for sorting ordinal categorical values, where their proposed method seems to outperform existing autoML approaches.

**Overall Review:**

- The authors should provide an additional ablation study on the necessity of the two approaches detailed in Section 3.2. The design detailed in Section 3.2 seems to indicate that the Bert-Sort algorithm is not even able to 1) sort [Hot, Boiling Hot, Lava Hot, Cold] into [Cold < Hot < Boiling Hot < Lava Hot] or 2) sort months according to their ordinal numbers, which seem to be extremely simple when a tremendously large model such as BERT is used. I think the author should provide experiments regarding Bert-Sort without using the two approaches mentioned above. Otherwise, I would presume it is the inductive bias in the first and the second approaches that induces the results in Table 4.

- In Table 5, I do not see a significant increase in the number of successful trained models when BERT-Sort is used. To be more specific, the number of successfully trained models is 23/50 w/ the BERT-Sort algorithm and 21/50 w/o the algorithm. Moreover, for AutoGluon, FLAML, and H2O, there is no impact when BERT-Sort is used.

**Reproducibility:**

The author provides detailed experimental procedures.

Yet, I am worried if the experiments are reproducible. Specifically speaking, it seems that the authors do not repeat their experiments over several random seeds. I am afraid that the results are cherry-picking.

**Review Confidence:**

3: You are fairly confident in your assessment. It is possible that you did not understand some parts of the submission or that you are unfamiliar with some pieces of related work.

**Review Rating:**

4: Marginally above the acceptance threshold (use sparsely)

**Review Summary:**

I like the ideas proposed by the authors, which I believe to be novel in the field of AutoML.
But I only recommend weak acceptance at this moment, as the empirical results are not extensive (i.e., do not provide ablation study), convincing (i.e., do not repeat over different random seeds), and significant (i.e., the results in Table 5 are far from being prominent).

---

### Official Review · Reviewer_6K9H · 2022-03-16

**Potential Impact On The Field Of Automl Rating:** 3
**Technical Quality And Correctness Rating:** 1
**Clarity Rating:** 3

**Summary Of Contributions:**

The authors' primary contribution is the development of a method of creating more meaningful feature embeddings for categorical variables that may carry (hidden) ordinal information. For instance, the words [mild, moderate, severe] make the most sense in terms of an ordinal relationship, and the authors posit that creating an embedding for this variable that respects this order could make ML models more successful at detecting relationships between these and other variables.

To enable these embeddings, the authors leverage MLM (masked language modeling) models, which are commonly used in NLP to predict a masked token in the context of other (unmasked) tokens. These predictions are then used to determine which ordering of the variables is most appropriate, in effect leveraging the MLMs' knowledge about what order these words appear in the corpora on which they are trained. The authors test many different choices for MLMs and emphasize that practitioners can choose from among many pre-trained models in the application of their technique.

Furthermore, the authors suggest that the method they use to determine the most likely ordering can be leveraged as a kind of metric to indicate whether a sequence of variables exhibits an ordinal relationship. This assertion, along with the primary one, is tested using different AutoML technologies, which the authors claim demonstrate the superiority of their solution over the default categorical embeddings found in these libraries.

**Clarity:**

On the easier side, the paper is in serious need of some proofreading. Some sections are better than others, but there is a general problem with subject-verb agreement and awkward phrasing that I found quite distracting. Some examples include: "One of the algorithms to process the context is Masked-Language Modeling (MLM), where it computes the probability..." (lines 98-99) and "Second, BERT-Sort instead of generating a large number of permutation cases when 𝜁 < |A|, it is sorting only 𝜁 number of elements, then it sorts the rest of elements one-by-one sequentially." (lines 147-148). In both of these cases, the meaning is clear but some work should be done to make the language more clear and natural.

Unfortunately, the part of the paper that suffered the most from a lack of clarity is the description of the main result of the paper. In the results in section 4.3, the results seem to be incongruous with the experiments as described. The authors use several data sets, which by their own description, include both classification and regression tasks. Yet all that is reported is the number of successes (the definition for which is never discussed nor can I imagine a definition I would accept) as well as F1 scores (which can be generalized to classification tasks but, as far as I know, make no sense for regression). The 10+% accuracy increase appears to be related to the increased F1 scores, which again are confusing and don't make much sense in the context of the problem.

I understood the authors' claim to be that their method increased the accuracy of an end-to-end AutoML model, but I don't see any results in the main paper nor the appendices that talk about the performance of a complete model. Instead, I see results about establishing the correct ordinal relationship of labels and these mysterious success/failure/F1 statistics.

**Overall Review:**

The authors have found a problem that many ML practitioners will find interesting and their solution seems to be a good one to me. I appreciate how the authors draw from the knowledge of NLP researchers to aid in other fields. The proposed solution is just complex enough to be interesting, while still being tractable to implement for any experienced ML user.

The paper needs a lot of attention to the language and technical aspects, however. The authors are unclear about what their main results are, which obscures from the reader whether they are meaningful. Mathematical and notational mistakes are common enough that they require the reader to make leaps to understand what the authors are saying in places. Finally, some results (e.g. the results about using MLMs as a metric of ordinal relation among tokens) seem to fail in spite of the text stating the opposite.

I would suggest that the authors carefully rewrite their paper with more clear results and some editing and resubmit at a future venue.

**Potential Impact On The Field Of Automl:**

It is possible that more meaningful unsupervised embeddings of variables would be of use to the AutoML community, as well as the ML community at large. As is discussed below, some technical issues and massive issues with clarity make it very difficult to know whether the impact cited in the abstract (10+% performance gain) is to be believed.

**Reproducibility:**

The model/method seems carefully explained to me and I believe that I could recreate their construction and use it as they used it. The lack of clarity mentioned in previous sections would keep me from being able to reproduce the final results, however.

I do not believe that I have enough information to run their experiments to verify their results as it stands.

**Review Confidence:**

3: You are fairly confident in your assessment. It is possible that you did not understand some parts of the submission or that you are unfamiliar with some pieces of related work.

**Review Rating:**

3: Marginally below the acceptance threshold (use sparsely)

**Review Summary:**

Due to the clarity issues and the many technical errors and typos throughout the paper, I do not recommend the paper be published in its current state. The idea seems promising and I believe the authors could have made a significant contribution to the field, but in its current state, it fails to deliver on its promises or convince me that it was performed thoroughly and competently.

**Technical Quality And Correctness:**

There are some glaring issues with correctness in this paper. I have broken them down into different topics as I experienced them:

**Complexity computations**

On lines 156-157 the authors say that their proposed computational method gets a speedup from $\mathcal{O}(n!)$ to $\mathcal{O}(\zeta\log(n))$ (plus a constant that doesn't affect the big-oh computation). This is highly dubious: consider the case when one chooses $\zeta=1$. Then the algorithm has to evaluate $1 + 2 + 3 + \cdots + n=\frac{n(n-1)}{2}$ different possible orderings, resulting in a runtime of $\mathcal{O}(n^2)$ if we assume that $\zeta$ is fixed. My back-of-the-envelope computations says the time complexity should be closer to $\mathcal{O}(n^\zeta + n^2)$ as long as $\zeta$ is chosen to be less than $\frac{n}{2}$. This still represents a speedup over the naive approach, but it is nowhere near logarithmic in $n$.

**Math notation**

There are a few math notational errors, including $U(M_{c,k})\exists \mathcal W_\eta,$ where the authors intended to write $U(M_{c,k})\in \mathcal W_\eta$. The $U$ notation, which may be common notation in NLP circles, was unfamiliar to me and I had to try to decode its meaning from context. Some clarification would help there. The notation $\hat P(M_{c,k}|S_c,\eta)$ was a bit misleading to me since it appeared to be computing the probability of a particular mask instead of the unmasked token which was masked in the application of the $k^{th}$ mask to the $c^{th}$ sequence. I don't think this was necessarily wrong as much as it was confusing. Some work on notation would make this whole section more legible.

**Algorithm 1**

There are a couple of small errors/typos that need to be addressed. The first occurs on line 16: here one is determining the new sequences to consider but it is computed as the union of $\mathcal C$, the collection of all $\zeta$ length combinations in $\mathcal A$, and $Seq(\mathcal C, \mathcal E)$, which I presume is supposed to denote the sequences created by inserting $\mathcal E$ at different places, as can be seen in Figure 2. I do not believe that you want to consider the sequences in $\mathcal C$ (these have already been computed in the first steps) and a better notation for what you want should be $Seq(\mathcal I,\mathcal E)$, the set of sequences created from inserting the token $\mathcal E$ into $\mathcal I$ at different places.

Later, on line 28, the authors use a union between what I understand to be a set (of probability values) and a single probability value. Set brackets around the latter value would aid in clarity although the intention can be deciphered. Since the values are summed at the end anyways, I would suggest the authors use an accumulator variable, simply adding the values after computing each.

**Ordinal value detection**

The results shown in the confusion matrix in Table 1 are quite bad. I computed their statistics again and got a precision of 0.875 (as stated in the paper), a recall of 0.204 (much different than the stated value of 1 in the paper), a specificity of zero, and an F1 score of 0.331 (which was claimed to be 0.933). These show either a severe miscalculation on the part of the authors or errors in transcribing their results. My takeaway is that the results as recorded demonstrate that the ordinal value detection using BERT-Sort fails horribly on its assigned task, with a large majority of sequences being misclassified.

**Section 4.2**

This section has a significant error. The formula for $Ord_{Acc}$ in equation (2) does not compute the value that appears in Table 3. The formula given is effectively a normalized version of Manhattan distance between two vectors (which also somewhat calls into question whether this is truly a novel metric), but the values shown in the table bear no resemblance to the values that come from using it. Worse, the definition given is a *metric* meaning that identical vectors will get a value of zero, instead of 1, which one would expect with a measurement of accuracy. I suspect the problem is that the definition in (2) is just wrong, but I cannot determine what the correct formula should be from the rest of the paper.

**Section 4.3**

There is a serious lack of clarity here (discussed in the next section), but from my understanding, the methodology is also questionable. I am assuming here that the intention of the authors (as it seemed from reading

---

### Official Review · Reviewer_61b6 · 2022-04-04

**Potential Impact On The Field Of Automl Rating:** 3
**Technical Quality And Correctness Rating:** 1
**Clarity Rating:** 3

**Summary Of Contributions:**

The authors introduce Bert-Sort, a novel method to semantically encode categorical values with zeros-hot Masked Language Models for tabular data. They compare their approach to existing techniques used in modern AutoML systems both in terms of accuracy of ordinal encodings and end-to-end AutoML performance via 10 datasets and show improvement in both areas.

**Clarity:**

Overall the paper is clear and well written. My major issues come with the comments I mentioned in technical quality & correctness, most importantly in Table 5.

**Overall Review:**

The authors propose a novel and interesting idea to leverage pre-trained language models for more accurate ordinal encoding in tabular datasets. This has strong potential to improve end-to-end AutoML systems by improving the quality of the data fed to downstream models. In particular, this problem has no easy solution outside of naive approaches as described by the authors currently implemented in AutoML systems, so leveraging pre-trained models to order the values is a step in the right direction, and shows encouraging results via Table 4.

However, I cannot ignore the major issues in Table 5 as I discuss in detail in earlier sections. Unfortunately this drastically hinders the paper due to a combination of misleading numbers, lack of information on reasons for dataset failures, a lack of overall per-dataset scores, a lack of reproducible code to run the experiments, and an incorrect conclusion that fails to discuss any drawbacks to the method or inconclusive findings from the experiments.

Because of these issues, I cannot accept the paper in its current form.

**Potential Impact On The Field Of Automl:**

Being able to correctly preprocess features in a fashion which maximizes predictive power is of high importance to AutoML systems, and ordinal encoding has been historically difficult to automate without hints from the user. Because this work mainly applies to a particular type of feature within Tabular data, this work is of medium impact.

**Reproducibility:**

The reproducibility checklist is filled out in the code repo and is mostly reasonable. However I do not see any clear links to code to reproduce the experiments beyond a written explanation for how they were conducted.

1. There is no mention of the versions of AutoML packages used in Figure 5 and the versions of the ML models used in Figure 9 in both the paper and provided code repo.
2. There is no code that I could find provided by the authors for actually running the experiments in Table 5. The code only provided the datasets in both original and post BERT-Sort transformation, but no training code.
3. No explanation of dataset failures.
4. No mention that I could find for the machines used to run the experiments beyond the training time. For example, no mention of CPU cores, memory, python version, etc.

**Review Confidence:**

5: You are absolutely certain about your assessment. You are very familiar with the related work and checked all the details carefully.

**Review Rating:**

3: Marginally below the acceptance threshold (use sparsely)

**Review Summary:**

This is a good paper hindered by a flawed experimental result analysis and conclusion which unfortunately cannot be overlooked in its severity. Because of these issues, I vote to reject. Refer to the other sections for detailed comments.

[Update from Rebuttal] I have increased my score from 2 to 3 due to the improvements made in the rebuttal.

**Technical Quality And Correctness:**

While I have only minor issues with the approach, theory, and experiments in Table 4 comparing against the OrdinalEncoder for Ordinal Accuracy & Accuracy, I do have severe issues with Table 5 and the conclusions found within.

1. L140 - How is the common word calculated? What if a value contains multiple common words? The implementation is not clear.
2. L156 - I am having trouble understanding how this could be O(Llog(n)) when n elements need to be passed to BERT Sort(cases). To me, this appears to be at minimum O(n^2). Additionally O(Llog(n)) implies that we could simply have L == n and get O(nlog(n)) instead of O(n!), which doesn't seem right.
3. L226 - "All AutoML tools failed on two data sets (regressions)" -> This is never explained as to why.
4. L226 - "and success on 8 classification problems" -> This is only true for H2O.
5. Table 5 - This table and the conclusions surrounding it are **extremely** misleading.

- Because you count failures as having a score of 0, the "10.61% performance gain" is purely due to the fact that TPOT and MLJar succeeded on more datasets with the BERT-Sort input. **Removing TPOT and MLJAR due to this issue, the conclusion would be that performance was made worse by BERT-Sort** due to the major drop in performance of H2O. **Correcting this issue completely flips the conclusion of the paper, yet the authors at no point mentioned this**.
- No explanation is ever made about what datasets each framework failed on and why, nor any investigation into the cause of H2O performing worse with BERT-Sort. Because this method doesn't change the format of the data, it is hard to imagine why BERT-Sort would improve the success rate of AutoML frameworks, which makes this extra confusing.
- Another issue is that for example it is impossible to tell if the 3 datasets succeeded by FLAML in Raw and Bert-Sort are even the same datasets, making the comparison of scores between them meaningless. There is no overall table that shows scores for each framework for each setting on each dataset.
- The choice to use F1-score is odd and was never explained, as it skews the importance of positive and negative classes. ROC-AUC and Acc both seem like more reasonable options, although this is a minor issue compared to those I mentioned previously.

---

### Official Review · Reviewer_ARU2 · 2022-04-06

**Potential Impact On The Field Of Automl Rating:** 2
**Technical Quality And Correctness Rating:** 2
**Clarity:** The paper is overall clear enough to …
**Clarity Rating:** 3

**Summary Of Contributions:**

This paper considers preprocessing of categorical features by treating them as text, to which a pretrained Transformer like BERT is applied to identify what would be the most likely ordering of these words in a natural corpus. Then the categories are ordinally encoded according to this ordering.



**Overall Review:**

This paper introduces a very simple idea with promising looking results. However, there are not enough comparisons/explanations of the details to convince me their results will be as useful in real-world AutoML as presented in the paper.

First off, there should be more baselines. Eg. one option is just to use pretrained (or fine-tuned) BERT to embed the categories' text into a higher-dimensional vector which can be fed into AutoML systems. This is considered in the following paper which should be discussed:

Benchmarking Multimodal AutoML for Tabular Data with Text Fields (2021)
https://arxiv.org/abs/2111.02705

Another option is to represent categories in a label-dependent way via target mean encoding. A third option is to use a feedforward network for the dataset with embedding layers for each categorical variable that learn embeddings of its categories from scratch. These embeddings could then easily be fed into the AutoML system.


Other questions:

- For reproducibility, what versions of each AutoML tool are you using, what machine are you running them on, are you running each tool as the only process on the machine?

- How much extra time does BERT-Sort add to the overall training of each AutoML system?

- Why did you choose F1 score as the predictions' evaluation metric?  I would be curious if the BERT-Sort gains remain across other evaluation metrics like AUC, accuracy.

- Why are there so many AutoML failures on your chosen datasets? Are you running on a machine with enough RAM, or are these datasets peculiar in some fashion?

- How do you handle missing values in your approach?

- How do you handle previously unseen category in the test data during inference?

**Potential Impact On The Field Of Automl:**

The authors show their BERT-Sort is reliably able to improve the performance of 4/5 AutoML systems over 10 datasets, compared to the default data preprocessing scheme used in those systems. Thus there could be moderate impact on the field of AutoML if these results generalize across more AutoML systems and datasets. That said, the authors did not compare with other preprocessing strategies, which perhaps might work as well as BERT-Sort for these particular datasets.

**Reproducibility:**

 For reproducibility, what versions of each AutoML tool are you using, what machine are you running them on, are you running each tool as the only process on the machine?

**Review Confidence:**

4: You are confident in your assessment, but not absolutely certain. It is unlikely, but not impossible, that you did not understand some parts of the submission or that you are unfamiliar with some pieces of related work.

**Review Rating:**

3: Marginally below the acceptance threshold (use sparsely)

**Review Summary:**

The presented idea is very simple the implement and can be combined with any AutoML system (although the authors should comment more on its additional computational complexity).  My biggest concern is around the trustworthiness and generalizability of the results (in particular the boost to AutoML systems' accuracy which seems like the major contribution of this paper).  If the authors can assuage my concerns, I am open to increasing more score, as the results do look promising and I like the simplicity.

**Technical Quality And Correctness:**

The way each AutoML tool was run seems correct to me, assuming you ran each tool on a separate machine with no other processes running. on the machine while the AutoML tool was running?

The results in Table 5 seem biased. If I understand correctly, you are giving an AutoML tool a F1 score = 0 when it fails on a dataset?  That seems like an overly strong penalty to me and muddies the comparison across different tools since their scores are over different datasets.  Also any intuition why BERT-Sort helps increase the number of successes for MLJAR and TPOT?

A minor quibble is you are not accounting for the additional time taken by BERT-Sort when comparing against the Raw run of each AutoML system.

---

### Official Review · Reviewer_J7ke · 2022-04-10

**Potential Impact On The Field Of Automl:** N/A for reproducibility reviewers
**Potential Impact On The Field Of Automl Rating:** 3
**Technical Quality And Correctness:** N/A for reproducibility reviewers
**Technical Quality And Correctness Rating:** 3
**Clarity:** N/A for reproducibility reviewers
**Clarity Rating:** 4

**Summary Of Contributions:**

This paper devises an approach to semantically encode ordinal categorical features as numerical features by using pre-trained language models when preprocessing tabular data, named BERT-Sort. In addition, this paper introduces a new benchmark that is collected from 10 different public data sets with 42 different ordinal features. Upon this benchmark, BERT-Sort significantly outperforms traditional encoding methods such as OrdinalEncoder.

**Ethics Details (Optional):**

N/A for reproducibility reviewers

**Overall Review:**

N/A for reproducibility reviewers

**Reproducibility:**

The reproducibility list is properly filled out and can be used to reflect the reproducibility of the proposed work.

Complete artifacts including the results shown in the paper and also intermediate results are released in the anonymized Github repository. However, it would be better if the authors could release the code as well. So our reviewers and other researchers would be able to reproduce the results and also apply this interesting BERT-Sort method to other tabular datasets.

**Review Confidence:**

4: You are confident in your assessment, but not absolutely certain. It is unlikely, but not impossible, that you did not understand some parts of the submission or that you are unfamiliar with some pieces of related work.

**Review Rating:**

5: Accept, good paper

**Review Summary:**

N/A for reproducibility reviewers

---

### Meta-Review · Area_Chair_qWmC · 2022-05-09

**Recommendation:** Accept
**Confidence:** 3

**Metareview:**

This paper considers preprocessing of categorical features by treating them as text, to which a pretrained Transformer like BERT is used to identify what would be the most likely ordering of these words in a natural corpus.

I would like to thank the authors for actively addressing reviewers' comments or providing clarifications whenever it was needed. As a result, ARU2 and 61b6, and 96Ap have raised their score and are inclined to (weakly) accept the paper. However, one of the reviewers raised many clarity issues, technical errors, and typos and suggested a weak reject.

I would recommend assigning a shepherd to this paper to address the reviewer's 6K9H in terms of presentations and writing.

---

### Decision · Program_Chairs · 2022-05-13

Accept